# Light-regulated allosteric switch enables temporal and subcellular control of enzyme activity

Mark Shaaya[1], Jordan Fauser[1], Anastasia Zhurikhina[2], Jason E Conage-Pough[3,4,5], Vincent Huyot[1], Martin Brennan[1], Cameron T Flower[3,4,5,6], Jacob Matsche[1], Shahzeb Khan[1], Viswanathan Natarajan[1], Jalees Rehman[1,7,8], Pradeep Kota[9], Forest M White[3,4,5,6], Denis Tsygankov[2], Andrei V Karginov[1,7]*

[1]Department of Pharmacology and Regenerative Medicine, The University of Illinois at Chicago, College of Medicine, Chicago, United States; [2]Wallace H. Coulter Department of Biomedical Engineering, Georgia Institute of Technology and Emory University School of Medicine, Atlanta, United States; [3]The David H. Koch Institute for Integrative Cancer Research, Massachusetts Institute of Technology, Cambridge, United States; [4]Center for Precision Cancer Medicine, Massachusetts Institute of Technology, Cambridge, United States; [5]Department of Biological Engineering, Massachusetts Institute of Technology, Cambridge, United States; [6]Program in Computational and Systems Biology, Massachusetts Institute of Technology, Cambridge, United States; [7]University of Illinois Cancer Center, The University of Illinois at Chicago, Chicago, United States; [8]Division of Cardiology, Department of Medicine, The University of Illinois, College of Medicine, Chicago, United States; [9]Marsico Lung Institute, Cystic Fibrosis Center and Department of Medicine, University of North Carolina, Chapel Hill, United States

*For correspondence:
karginov@uic.edu

**Abstract** Engineered allosteric regulation of protein activity provides significant advantages for the development of robust and broadly applicable tools. However, the application of allosteric switches in optogenetics has been scarce and suffers from critical limitations. Here, we report an optogenetic approach that utilizes an engineered Light-Regulated (LightR) allosteric switch module to achieve tight spatiotemporal control of enzymatic activity. Using the tyrosine kinase Src as a model, we demonstrate efficient regulation of the kinase and identify temporally distinct signaling responses ranging from seconds to minutes. LightR-Src off-kinetics can be tuned by modulating the LightR photoconversion cycle. A fast cycling variant enables the stimulation of transient pulses and local regulation of activity in a selected region of a cell. The design of the LightR module ensures broad applicability of the tool, as we demonstrate by achieving light-mediated regulation of Abl and bRaf kinases as well as Cre recombinase.

## Introduction

Dissection of biological processes is greatly aided by optogenetic methods that allow researchers to mimic and manipulate the function of individual proteins. Multiple broadly applicable approaches have been developed to enable tight regulation of protein localization and protein interactions (*Guntas et al., 2015*; *Wang et al., 2016*; *Strickland et al., 2012*; *Kawano et al., 2015*); however, the direct control of enzymatic activity remains a challenge. Only a few existing strategies achieve direct regulation of enzymatic activity, but they either lack the ability to activate enzymes locally within a cell, or they are difficult to apply to multiple enzyme classes (*Wang et al., 2019*;

**eLife digest** Cells need to sense and respond to their environment. To do this, they have dedicated proteins that interpret outside signals and convert them into appropriate responses that are only active at a specific time and location within the cell.

However, in many diseases, including cancer, these signaling proteins are switched on for too long or are active in the wrong place. To better understand why this is the case, researchers manipulate proteins to identify the processes they regulate. One way to do this is to engineer proteins so that they can be controlled by light, turning them either on or off.

Ideally, a light-controlled tool can activate proteins at defined times, control proteins in specific locations within the cell and regulate any protein of interest. However, current methods do not combine all of these requirements in one tool, and scientists often have to use different methods, depending on the topic they are researching.

Now, Shaaya et al. set out to develop a single tool that combines all required features. The researchers engineered a light-sensitive 'switch' that allowed them to activate a specific protein by illuminating it with blue light and to deactivate it by turning the light off. Unlike other methods, the new tool uses a light-sensitive switch that works like a clamp. In the dark, the clamp is open, which 'stretches' and distorts the protein, rendering it inactive. In light, however, the clamp closes and the structure of the protein and its activity are restored. Moreover, it can activate proteins multiple times, control proteins in specific locations within the cell and it can be applied to a variety of proteins.

This specific design makes it possible to combine multiple features in one tool that will both simplify and broaden its use to investigate specific proteins and signaling pathways in a broad range of diseases.

*Dagliyan et al., 2016*; *Zhou et al., 2017*; *Diaz et al., 2017*; *Wu et al., 2009*; *Hongdusit et al., 2020*), thus limiting their use for addressing mechanistic questions in cell biology. This highlights the need for an optogenetic approach which is broadly applicable, easy to implement and provides all the advantages of optogenetics in one tool including tight temporal and local subcellular regulation. An attractive strategy that can be harnessed for this purpose is the application of a rationally designed light-sensitive domain that can allosterically control protein activity. This protein engineering approach offers several important advantages. It enables targeted regulation of one domain within a multidomain protein (*Karginov et al., 2010*). Regulation is achieved 'remotely' without steric interference with the catalytic pocket of the enzyme and substrate binding. Also, allosteric switch domains are genetically encoded into the targeted protein simplifying the application of the tool. Only four optogenetic methods for allosteric regulation of activity have been reported so far and they suffer from critical limitations (*Dagliyan et al., 2016*; *Wu et al., 2009*; *Hongdusit et al., 2020*; *Winkler et al., 2015*; *Reynolds et al., 2011*). Two of these strategies achieve subcellular control of certain targeted proteins but they cannot be applied to many other enzymes due to their design (*Wu et al., 2009*; *Hongdusit et al., 2020*). One approach is broadly applicable but it does not achieve local control and triggers inactivation rather than activation of the protein (*Dagliyan et al., 2016*). A similar approach described by Reynolds et al achieves regulation in both directions for dihydrofolate reductase (DHFR) and PDZ domain but efficiency of this approach for applications in mammalian cells and its ability to regulate proteins at subcellular level has not been demonstrated (*Reynolds et al., 2011*). Furthermore, existing allosteric switches lack the ability to tune the activation/inactivation kinetics, an important feature required for mimicking different temporal signaling modes of enzymes.

Among the various potential targets of value for cell biology applications, protein kinases constitute an important family of enzymes that regulate key physiological functions and thus their activity is tightly controlled. Aberrant kinase regulation is the underpinning of many diseases, including the development and progression of malignant tumors (*Lee and Yaffe, 2016*). Many kinases are therefore important therapeutic targets and significant research effort is directed towards uncovering their function in cells. Such functional studies are challenging because a single kinase often induces drastically different responses depending on the location, level and timing of its activation

(*Marshall, 1995*; *Rauch et al., 2016*; *Cohen-Saidon et al., 2009*; *Bugaj et al., 2018*; *Toettcher et al., 2013*). Furthermore, transient, sustained, or oscillatory kinase activation can result in distinct outcomes. Therefore, interrogating kinase-mediated signaling requires approaches capable of mimicking complex spatiotemporal control of kinase activity down to the subcellular resolution. Traditional methods using genetic manipulation and small molecule inhibitors lack this level of regulation. Optogenetics, on the other hand, overcomes these limitations by leveraging the precision of light-mediated activation (*Toettcher et al., 2011*). Some existing optogenetic tools regulate the dimerization or localization of kinases (*Toettcher et al., 2013*; *Wend et al., 2014*; *Grusch et al., 2014*; *Chang et al., 2014*; *Kim et al., 2014*). These approaches control the catalytic activity only indirectly and can only be applied to a narrow group of kinases. Other approaches employ light-sensitive caging or allosteric switches for the regulation of kinase catalytic domains (*Wang et al., 2019*; *Dagliyan et al., 2016*; *Zhou et al., 2017*). However, these methods do not achieve subcellular control of activity and some of these approaches are also limited in their broader applicability due to structural requirements (*Wang et al., 2019*; *Dagliyan et al., 2016*; *Zhou et al., 2017*). Thus, the development of a broadly applicable approach to enable optogenetic regulation of protein kinases remains challenging and highly desirable.

Here, we report the development of a protein engineering approach that enables light-mediated allosteric control of enzymatic activity and combines critical advantages of optogenetics in one tool: it provides tight temporal regulation of activity with tunable kinetics, enables local control that can be achieved with subcellular resolution, and allows for broad applicability to different types of enzymes. We achieve this by employing an engineered light sensitive switch domain that is genetically encoded and enables the allosteric regulation of kinase catalytic domains. It provides fast, specific, and tunable control of kinase activity in living cells using low intensity blue light. Furthermore, this method enables reversible and repetitive activation of a kinase and provides local control of kinase activity at a subcellular level. Application of this tool allowed us to uncover temporal dynamics of phosphorylation-mediated signaling following short term (10 s) as well as prolonged activation of a kinase. Importantly, we show that this approach can be applied not only to different kinases but also to other enzymes.

## Results

### Development of a Light-Regulated kinase

To modulate kinase activity with light, we engineered a Light-Regulated (LightR) domain that can potentially function as an allosteric switch when inserted into a catalytic domain of a kinase. LightR is comprised of two tandemly connected Vivid (VVD) photoreceptor domains from *Neurospora crassa* (*Nihongaki et al., 2014*; *Vaidya et al., 2011*; *Zoltowski and Crane, 2008*; *Zoltowski et al., 2007*). VVD is a monomer in the dark, and it forms an antiparallel homodimer upon illumination with blue light (*Nihongaki et al., 2014*; *Vaidya et al., 2011*; *Zoltowski and Crane, 2008*; *Zoltowski et al., 2007*; *Wang et al., 2012*). Dimerization is accompanied by a major flip of the N-terminal tail, bringing it close to the C-terminus of the other VVD in the dimer (*Figure 1A*; *Vaidya et al., 2011*; *Zoltowski and Crane, 2008*; *Zoltowski et al., 2007*). Therefore, we surmised that a tandem connection of two VVDs via a flexible linker would generate a clamp-like switch of 335 amino acid total size that opens in the dark and closes in response to blue light. To connect two VVD molecules, we designed a flexible 22 amino acid linker $(GGS)_4G(GGS)_3$ which provides sufficient flexibility and length (approximately 25–30 Å when extended in the dark state) to accommodate the association and dissociation of the VVD monomers. We hypothesized that inserting this LightR clamp domain into a small flexible loop within the catalytic domain of an enzyme would enable light-mediated regulation of its activity. In the dark, the opening of the LightR clamp could increase the distance between its N- and C- termini up to approximately 25 Å, which should distort the native structure of the catalytic domain and thereby inactivate the enzyme. Illumination with blue light would close the clamp and bring the N- and C-termini of LightR together resulting in restoration of the native structure of the catalytic domain and recovery of the enzyme activity (*Figure 1B*).

We first applied this optogenetic approach to regulate the tyrosine kinase Src which is a key regulator of cell migration, angiogenesis and tumor progression (*Sen and Johnson, 2011*), and generated light-regulated tyrosine kinase Src (LightR-Src). Our previous studies demonstrate that the

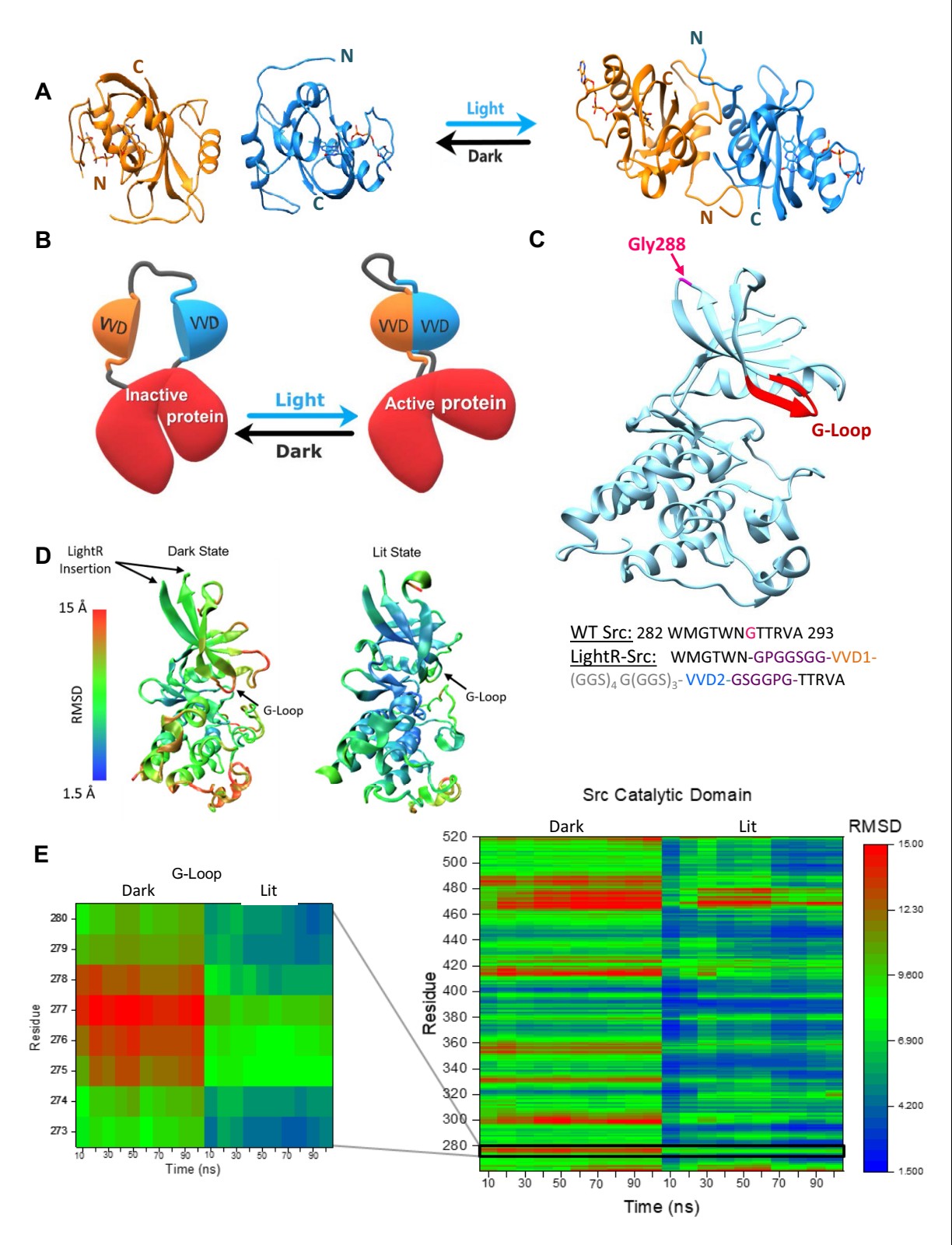

**Figure 1.** LightR-Src design and molecular dynamics simulations. (**A**) Crystal structures of two Vivid monomers in the dark state (PDB: 2PD7), and the dimer in the lit state (PDB: 3RH8). (**B**) Cartoon representation of LightR design. Two tandemly connected VVD photoreceptors inserted in the catalytic domain disrupt the catalytic activity of the protein in the dark. Dimerization of VVD in response to blue light restores the protein activity. (**C**) Crystal structure of c-Src catalytic domain (PDB:1Y57) with the insertion site G288 in magenta. The insertion site is connected to the catalytically important

*Figure 1 continued on next page*

*Figure 1 continued*

G-loop , highlighted in red, by a β-strand. Schematic below shows the amino acid sequence of the wild type Src residues around the insertion site and the resulting construct with LightR insertion. Insertion site G288 in WT Src is shown in magenta, asymmetric flexible GPGGSGG and GSGGPG linkers are depicted in purple, VVD proteins are shown in orange and blue, and 22-residue flexible linker is shown in grey. (D, E) Computational modeling of structural changes in the catalytic domain of LightR-Src. Color scale reflects the degree of deviation from the position in the crystal structure of Src (PDB: 1Y57). (D) Structural models reflecting the average RMSD of each residue for the dark and the lit states. (E) Comparative heat map of RMSD values for each residue over the course of the simulation for Src catalytic domain in lit and dark states. Zoomed-in insert shows changes in the G-loop. RMSD values represent an average from three independent 100 ns simulations for each state, lit and dark.

The online version of this article includes the following source data and figure supplement(s) for figure 1:

**Source data 1.** Supplementary source data for *Figure 1D,E*, and *Figure 1—figure supplements 1–3*.
**Figure supplement 1.** Comparison of RMSD values for residues in LightR-Src and wild type Src.
**Figure supplement 2.** Cross-correlation maps of the dark and lit state of LightR-Src.
**Figure supplement 3.** G-Loop displacement caused by the LightR switch in the dark.

insertion of a regulatory domain at position Gly288 in the Src catalytic domain can be employed to achieve allosteric control of activity *Karginov et al., 2010*; thus we focused on this site for the insertion of LightR domain (*Figure 1C*). We first performed molecular dynamics simulations of the designed LightR-Src catalytic domain to predict a possible mechanism for the regulation of kinase activity by LightR domain. To build the initial LightR domain model for the dark and lit state, we modified the crystal structures of VVD in the light (PDB: 3RH8) and dark (PDB: 2PD7) by connecting the VVD proteins with a 22-residue flexible linker, (GGS)$_4$G(GGS)$_3$. The GPGGSGG and GSGGPG linkers were then added to the N- and C-termini of the LightR domain, respectively (*Figure 1C*). All of these flexible linkers were refined in Modeller. The LightR-Src structures were then constructed by inserting LightR domain into crystal structures of c-Src catalytic domain (PDB:1Y57) replacing G288. To ensure the best possible approximation of a starting structure, the engineered molecules were also subjected to energy minimization to allow for resolution of potential steric clashes, unusual torsion angles, or other energetically unfavorable aspects potentially introduced in the construction of the model. The geometry optimized molecules were equilibrated at 37°C to reach an equilibrium state structure at the desired temperature prior to starting the simulations. The analysis of 100 ns simulations demonstrated stabilization of fluctuations in root mean square deviation (RMSD) of the backbone (*Figure 1—figure supplement 1A*). This is indicative of a structure occupying an energetic minima and thus represents a reasonable structure for further analysis. Analysis of LightR-Src in the dark shows increased RMSD of individual residues within the catalytic domain relative to the active wild-type Src (*Figure 1D,E*; *Figure 1—figure supplement 1B*) and a strong correlation between the motion of the residues in the LightR clamp and in the Src catalytic domain (*Figure 1—figure supplement 2*) suggesting a possible allosteric mechanism. In contrast, the lit state of LightR-Src exhibits much lower RMSD, closer to wild-type Src (*Figure 1D,E*; *Figure 1—figure supplement 1B*), and loses the motion correlation between LightR domain and Src residues (*Figure 1—figure supplement 2*). This indicates that the open conformation of LightR clamp in the dark can cause changes in the catalytic domain of Src that can be reversed upon illumination with blue light. Interestingly, LightR-Src in the dark shows substantial deviation of the G-loop (*Figure 1D,E*; *Figure 1—figure supplements 1B* and *3*), a critical functional element of most protein kinases (*Cowan-Jacob, 2006*; *Krupa et al., 2004*). This suggests a possible allosteric mechanism by which LightR domain regulates kinase activity via disruption of the G-loop.

To test whether LightR insertion enables light-mediated regulation of Src, we evaluated Src catalytic activity by an *in vitro* kinases assay using purified N-terminal fragment of paxillin as a substrate (*Karginov et al., 2010*; *Cai et al., 2008*; *Klomp et al., 2016*). In this and all following experiments, we used a LightR-Src construct bearing a Y527F mutation (avian cSrc position) that disrupts Src autoinhibition and prevents its negative regulation by endogenous mechanisms (*Karginov et al., 2010*; *Zheng et al., 2000*). We employed this to ensure that LightR-Src is only regulated by light. We compared LightR-Src activity to the activity of the catalytically inactive mutant of LightR-Src (D388R in mouse cSrc, sequence ID: NP_001020566.1) and the constitutively active mutant of cSrc (Y527F in avian cSrc). LightR-Src, as well as the control constructs, were transiently overexpressed in and immunoprecipitated from LinXE cells. Our results show that LightR insertion in Src completely disrupts its activity in the dark which is recovered in response to blue light to levels comparable to that

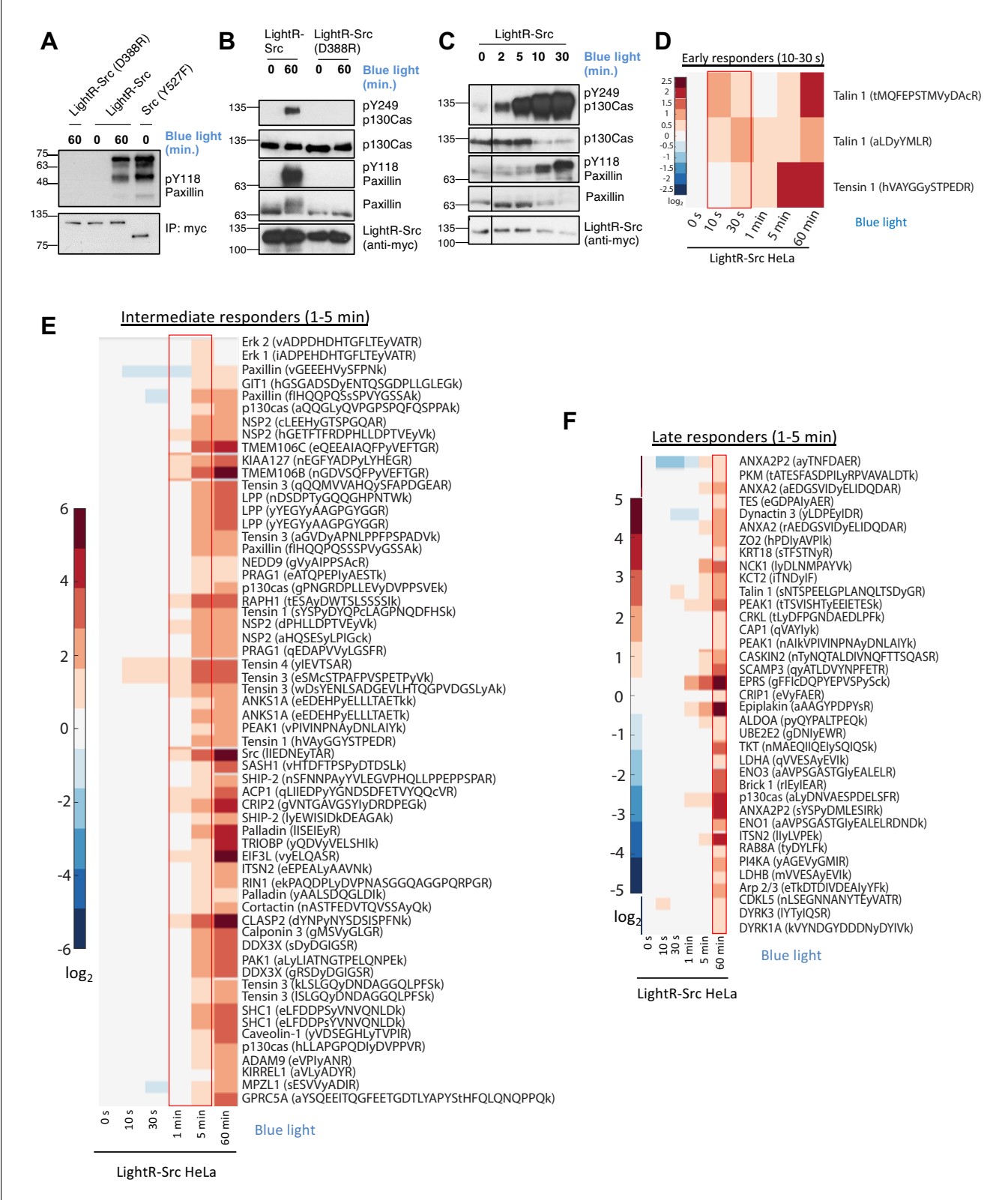

**Figure 2.** Characterization of LightR-Src. (**A**) Analysis of LightR-Src using *in vitro* kinase assay. LinXE cells transiently expressing the indicated Src constructs bearing an mCherry and a myc tag at the C-terminus were exposed to continuous blue light for 60 min where indicated. Src constructs were immunoprecipitated and their ability to phosphorylate purified N-terminal fragment of paxillin was assessed using an *in vitro* kinase assay. (**B, C**) Phosphorylation of endogenous paxillin and p130Cas in response to activation of LightR-Src in living cells. LinXE cells transiently expressing indicated

*Figure 2 continued*

LightR-Src construct bearing a tandem mCherry-myc tag at the C-terminus were continuously illuminated with blue light for the specified periods of time. Cell lysates were probed for phosphorylation of Src substrates, paxillin and p130Cas. All experiments were repeated at least three times with similar results. (D-F), Phosphoproteomics analysis of temporal changes in protein phosphorylation induced by LightR-Src activation in HeLa cells stably expressing LightR-Src-mCherry-myc. Heatmaps represent three categories defined by the initial time of upregulation: Early responders (D, 10–30 s), Intermediate responders (E, 1–5 min), and Late responders (F, 1 hr). Phosphopeptides are clustered using the correlation distance metric. Columns represent relative abundances of a phosphopeptide at given timepoints normalized to 0 s. Data shows average of three independent experiments. The online version of this article includes the following source data and figure supplement(s) for figure 2:

**Source data 1.** Supplementary source data for *Figure 2D–F* and *Figure 2—figure supplements 2–4*.
**Figure supplement 1.** Characterization of LightR-Src regulation and expression.
**Figure supplement 2.** Principal component analysis and STRING biological network analysis of changes in phosphoproteome caused by activation of LightR-Src.
**Figure supplement 3.** Transient phosphorylation of ERK kinases induced by activation of LightR-Src.
**Figure supplement 4.** Phosphoproteomic analysis of temporal changes in protein phosphorylation for control HeLa cells exposed to blue light for the indicated times.

of the constitutively active Src (*Figure 2A*). Next, we tested the regulation of LightR-Src in living cells by evaluating changes in phosphorylation of known endogenous Src substrates, p130Cas (Y249) and paxillin (Y118) (*Cunningham-Edmondson and Hanks, 2009*; *Sachdev et al., 2009*). Our results show that illumination of LinXE cells, transiently expressing LightR-Src, induces robust phosphorylation of the endogenous Src targets paxillin and p130Cas (*Figure 2B*, *Figure 2—figure supplement 1A*). Importantly, cells expressing the catalytically inactive mutant of LightR-Src (D388R) that were also exposed to blue light did not show any increase in phosphorylation of Src substrates (*Figure 2B*). This demonstrates that the increase in phosphorylation is a direct consequence of LightR-Src activation and that blue light by itself had no effects on the Src targets. The activation time-course demonstrates noticeable phosphorylation of Src targets after only two minutes of light irradiation (*Figure 2C*). The level of activity of LightR-Src can be regulated by attenuation of light intensity (*Figure 2—figure supplement 1B*). These data support our model for regulation of kinase activity using LightR clamp domain and demonstrate efficient and specific regulation of LightR-Src in living cells.

To further assess the function of LightR-Src and the kinetics of its activation in cells, we evaluated the phosphorylation of a broad panel of Src substrates at different time points of activation using quantitative LC-MS/MS analysis. For this analysis, we used HeLa cell line stably expressing LightR-Src-mCherry-myc construct at levels comparable to that of endogenous Src (*Figure 2—figure supplement 1C*). Phosphorylation of known Src targets, including caveolin1 (Y14), p130Cas (Y12, Y128, Y249, and Y287), paxillin (Y88 and Y118), p120 catenin (Y217 and Y228), and cortactin (Y421), was significantly increased in LightR-Src expressing cells, but not in control cells, exposed to blue light (*Figure 2—source data 1*). Principal component analysis (PCA) revealed that LightR-Src-expressing cells exhibited a broadly distinct phosphoproteome dynamics from control cells following light exposure (*Figure 2—figure supplement 2A*). Protein interaction network analysis along principal component one revealed that LightR-Src-induced phosphorylation events were enriched for cell migration, cell adherens junctions, and focal adhesions, all of which are processes known to be driven by Src activation (*Figure 2—figure supplement 2B*). To assess the kinetics of LightR-Src signaling, we analyzed changes in the phosphoproteome at different time points after LightR-Src activation. Several distinct phosphorylation kinetics profiles were identified. 'Early responders' exhibited a significant increase in phosphorylation as early as ten to thirty seconds after LightR-Src activation, further demonstrating rapidity of activation achieved by LightR-Src (*Figure 2D*). 'Intermediate responders' show increased phosphorylation at the 1 to 5 min time points (*Figure 2E*). Interestingly, we detected Src autophosphorylation on Y416 at 1 min, indicating that Src phosphorylates some targets before it even undergoes autophosphorylation. A group of 'late responders' were phosphorylated at 1 hr (*Figure 2F*). A distinct group of proteins comprised of MAP kinases ERK1 and ERK2 exhibited only transient increase in phosphorylation at 5 min (*Figure 2—figure supplement 3*). This phosphorylation is mediated by upstream MAP kinase cascades and leads to activation of ERK kinases (*Bugaj et al., 2018*; *Toettcher et al., 2013*; *Cargnello and Roux, 2011*). Thus, our data indicate that Src only transiently activates specific MAP kinase pathways. Importantly, all these phosphorylation changes were not detected in control HeLa cells that were exposed to blue light but did not

express LightR-Src (*Figure 2—figure supplement 4A–C*). Overall, these results demonstrate that LightR-Src phosphorylates known Src substrates, show the fast kinetics of LightR-Src signaling within seconds, and uncover distinct temporal patterns of Src target protein phosphorylation in living cells.

Since VVD dimerization is reversible (*Kawano et al., 2015*), we hypothesized that LightR-Src should become inactive when light is switched off. To test this, LinXE cells transiently expressing LightR-Src were illuminated with blue light for 30 min and then placed in the dark for different periods of time. Our results show that incubation in the dark led to a significant decrease in phosphorylation of paxillin (*Figure 3A*). However, it took up to 2 hr for phosphorylation to return to basal levels, indicating the slow inactivation kinetics of LightR-Src. To achieve faster inactivation of LightR-Src, we introduced an I85V mutation into both VVD domains. This mutation reduces the half-life of VVD dimer in the dark from 18,000 s to 780 s and thus should facilitate faster LightR-Src inactivation (*Zoltowski et al., 2009*). Indeed, our results demonstrate that compared to the original LightR-Src, the I85V/I85V variant (FastLightR-Src) shows much faster reversibility. Within two minutes after the light was switched off, we observed a significant decrease in paxillin phosphorylation (*Figure 3B*). By fifteen minutes, phosphorylation reached basal level. However, we noticed that activation of FastLightR-Src leads to a lower p130Cas phosphorylation level when compared to the same activation time point of LightR-Src (*Figure 3—figure supplement 1*). This is potentially due to the fast cycling of I85V mutants between lit and dark state (*Zoltowski et al., 2009*). This cycling could happen even when cells are illuminated and thus would reduce the fraction of active LightR-Src molecules at a given time (*Zoltowski and Crane, 2008*). Thus, FastLightR-Src may allow researchers to mimic function of Src kinase cycling between activation and inactivation states in living cells (*Kaimachnikov and Kholodenko, 2009*). Overall, our results show that the off-kinetics of the LightR switch can be tuned by modifications of the VVD domains. This provides the flexibility required for mimicking different temporal modes of kinase signaling, a capability that existing optogenetics approaches lack (*Wang et al., 2019*; *Dagliyan et al., 2016*; *Zhou et al., 2017*).

Protein kinases are often activated transiently and can undergo repeated cycles of activation/inactivation (*Kholodenko, 2006*; *Purvis and Lahav, 2013*; *Conlon et al., 2016*; *Li et al., 2017*; *Zhang et al., 2018*; *Roche et al., 1995*; *Hilioti et al., 2008*; *Jacquel et al., 2009*; *Kholodenko, 2000*; *Maeda et al., 2004*). These oscillations of kinase activity can drive a specific biological response (*Kholodenko, 2006*; *Purvis and Lahav, 2013*; *Conlon et al., 2016*; *Li et al., 2017*; *Roche et al., 1995*; *Hilioti et al., 2008*; *Jacquel et al., 2009*; *Kholodenko, 2000*; *Maeda et al., 2004*). Thus, we wanted to determine whether FastLightR construct can be used to mimic oscillations of kinase activity in living cells. To test this, FastLightR-Src was activated for two periods of

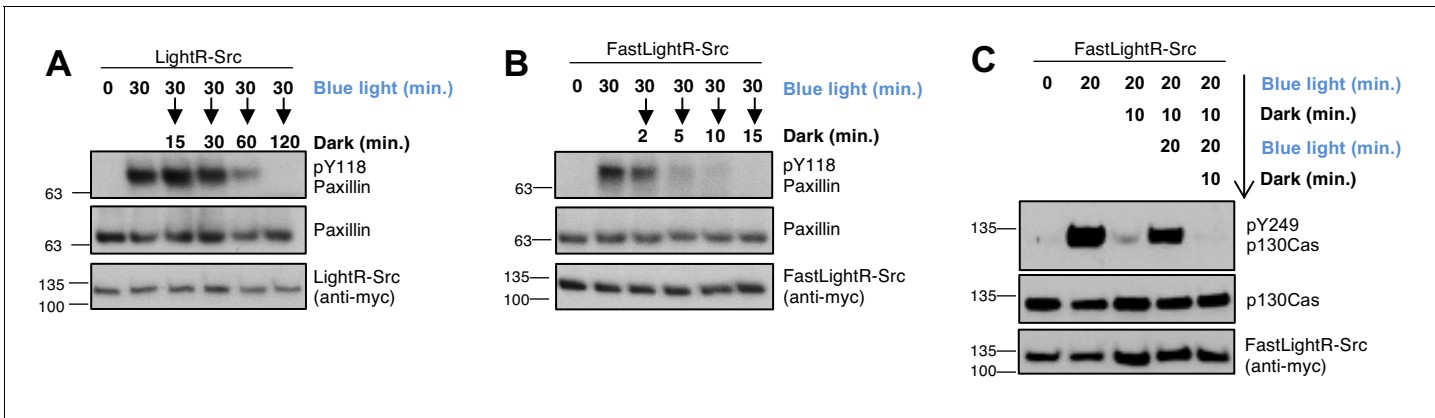

**Figure 3.** Reversibility of LightR- and FastLightR-Src. LinXE cells transiently expressing the indicated Src constructs bearing tandem mCherry-myc tag at the C-terminus were continuously exposed to blue light for specified times and then either placed in the dark for different periods of time (**A, B**) or repeatedly incubated in the dark for 10 min and in the light for 20 min (**C**). Cell lysates were collected and probed for phosphorylation of Src substrates. All experiments were repeated at least three times with similar results.

The online version of this article includes the following figure supplement(s) for figure 3:

**Figure supplement 1.** Comparison of LightR-Src and FastLightR-Src activity.

twenty minutes each, separated by ten minutes of deactivation. Our results reveal successful cycles of activation and inactivation as indicated by changes in phosphorylation of p130Cas (*Figure 3C*).

Previous studies show that activation of Src leads to stimulation of cell spreading (*Klomp et al., 2016*; *Karginov et al., 2014*; *Kaplan et al., 1995*; *Cary et al., 2002*; *Fu et al., 2018*). Therefore, we tested whether LightR-Src activation induces a similar response in living cells. Our results show that cells expressing FastLightR-Src start spreading upon irradiation with blue light (*Figure 4A*; *Video 1*) and stop immediately when the light is turned off. Repeated irradiation of cells with blue light induced corresponding cycles of cell spreading; demonstrating again that the LightR tool can be used to mimic oscillation of kinase activity in living cells (*Figure 4A*). Importantly, illumination of cells expressing catalytically inactive mutant of LightR-Src (D388R) did not induce any cell-spreading (*Figure 4B*). Also, we observed that inactive FastLightR-Src localizes in the perinuclear region and translocates to focal adhesions and cell membrane upon activation (*Figure 4C*; *Video 1*). This translocation is reversible and correlates with activation/inactivation cycles. Notably, this change in localization mimics that of wild type Src (*Kaplan et al., 1995*; *Chu et al., 2014*). Overall, our results demonstrate that LightR-Src activation can mediate cell morphodynamic changes and functions similarly to what has been observed for native Src kinase.

## Subcellular activation of LightR-Src

Current optogenetic tools enable localized kinase signaling only by re-localizing and sequestering a constitutively active kinase to an organelle or to specific areas in the cell (*Kerjouan, 2019*; *O'Banion et al., 2018*; *Kakumoto and Nakata, 2013*). Light-mediated regulation of kinase catalytic activity per se has not been achieved at a subcellular level. We hypothesized that FastLightR-Src activation/inactivation kinetics should enable local activation of the kinase only in a defined subcellular location. Localized activation of protein kinases and Src is a critical determinant in the regulation of cell function. Previous studies have suggested that local activation of Src at the cell periphery should stimulate the formation of local membrane protrusions (*Cary et al., 2002*; *Baumgartner et al., 2008*). However, this hypothesis has only been indirectly supported and has not been rigorously evaluated due to the limitations of existing methods. The LightR-Src approach would allow us to define the effects of local activation of Src in living cells. Indeed, we observed that local illumination of HeLa cells transiently expressing FastLightR-Src induced the formation of membrane protrusions within the illuminated area and caused polarization of the cell towards the light (*Figure 5A–C*; *Figure 5—figure supplement 1A*; *Video 2*). This effect was reversed as soon as the light was switched off (*Figure 5A*; *Video 2*). Cells expressing the catalytically inactive LightR-Src (D388R) did not polarize in response to local light irradiation (*Figure 5B,C*). Notably, FastLightR-Src translocated to focal adhesions only in the area illuminated with blue light (*Figure 5D*; *Figure 5—figure supplement 1B*; *Video 2*) and relocated back once the light was switched off (*Figure 5D*; *Video 2*). This is again consistent with known activation-dependent changes in localization of wild type Src (*Dhar and Shukla, 1991*; *Weernink and Rijksen, 1995*). These results reveal that local activation of Src is sufficient to induce local protrusions and demonstrate that LightR approach can be used for the regulation of kinase activity at a subcellular level.

Local regulation of protein kinase allows us to assess the dynamics of local morphological changes. A previous study using Src family kinases (SFK) biosensor suggested strong correlation between SFK activity at the cell periphery and cell-edge velocity (*Gulyani et al., 2011*). LightR-Src allows us to determine the direct effect of local Src activity on protrusion dynamics. To achieve this, we evaluated temporal changes in cell edge velocity upon continuous local activation of LightR-Src in HeLa cells. Our analysis revealed that Src induced local waves of increased cell-edge velocity (*Figure 5E,H*; *Figure 5—figure supplement 2A*); suggesting the induction of recurrent local contractions that slow down membrane protrusion. Inhibition of Rho associate protein kinase (ROCK) and myosin light chain kinase (MLCK), major regulators of contractile machinery (*Parri et al., 2007*; *Totsukawa et al., 2000*), perturbed these waves and reduced the average velocity of protrusions (*Figure 5E–J*; *Figure 5—figure supplement 2B*). Interestingly, inhibition of ROCK caused significantly smaller average velocity reduction than inhibition of MLCK (*Figure 5—figure supplement 2B*), consistent with its limited role in phosphorylating MLC at the cell periphery (*Totsukawa et al., 2004*). Thus, by applying LightR-Src, we demonstrated that sustained local Src activity induces waves of protrusions that are mediated by ROCK and MLCK.

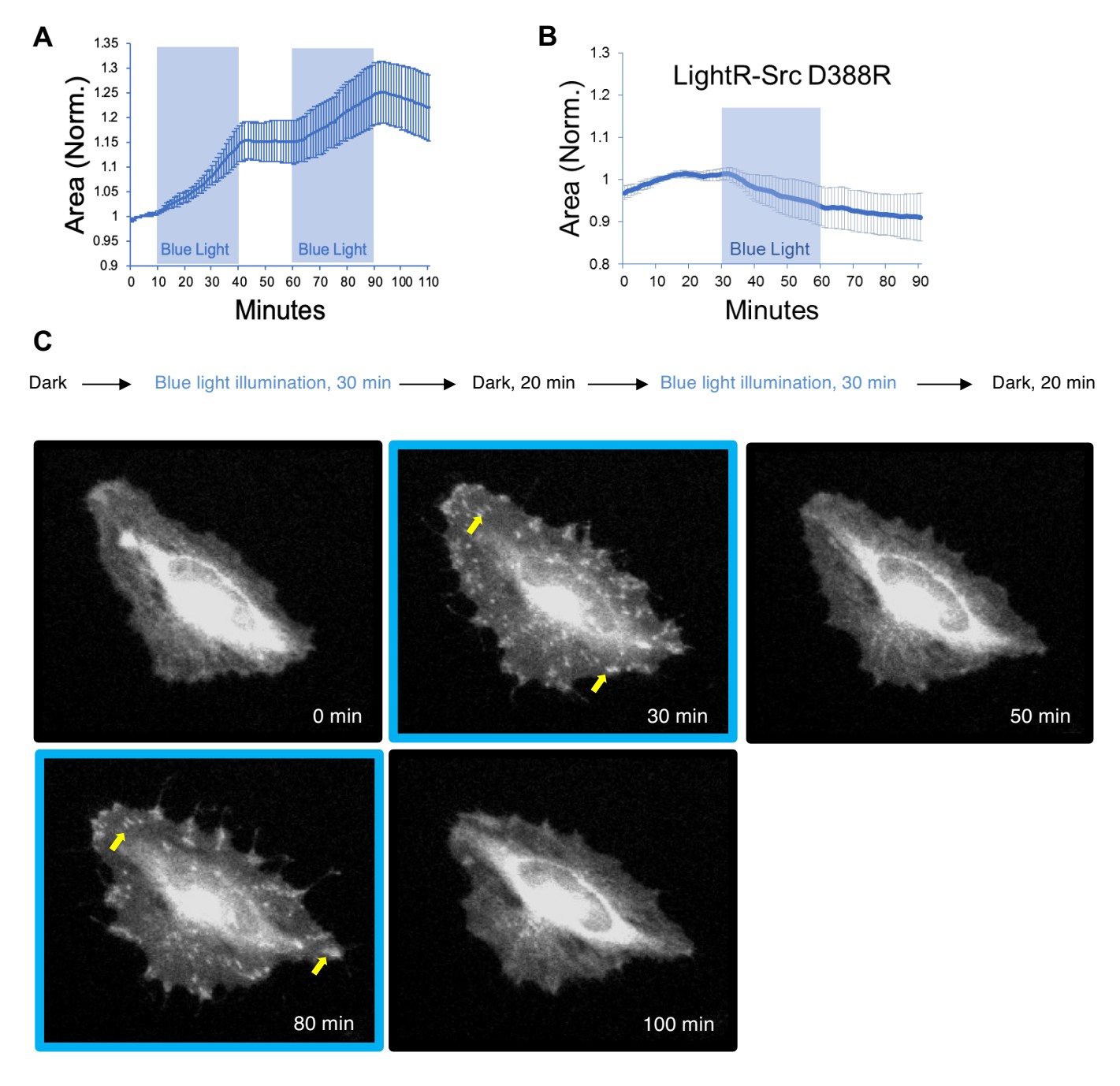

**Figure 4.** Regulation of cell morphology and LightR-Src localization by light. HeLa cells transiently co-expressing FastLightR-Src-mCherry-myc (N = 10 cells) or catalytically inactive LightR-Src (D388R)-mCherry-myc (N = 9 cells) with stargazin-iRFP670 (plasma membrane marker) were imaged live every minute while illuminated for indicated periods of time (blue rectangles). (A, B) Quantification of changes in cell area induced by activation of LightR-Src. Graphs represent mean ±90% confidence intervals. (C) Representative images of a HeLa cell showing changes in LightR-Src localization upon illumination with blue light (see *Video 1*). Yellow arrows point to FastLightR-Src localization at structures resembling focal adhesions.

The online version of this article includes the following source data for figure 4:

**Source data 1.** Supplementary source data for *Figure 4A*.
**Source data 2.** Supplementary source data for *Figure 4B*.

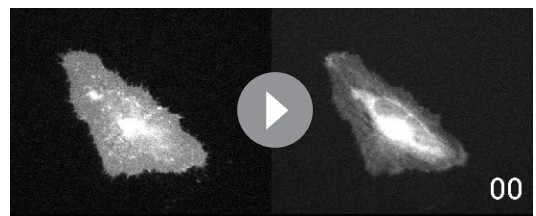

**Video 1.** Time-lapse of HeLa cell co-expressing stargazin-iRFP (membrane marker, left panel) and FastLightR-Src-mCherry (right panel) imaged every minute using widefield fluorescence microscopy. Cell was globally illuminated with blue light (50 ms pulses every second, 50 pulses per minute) as indicated. Time displayed in minutes.
https://elifesciences.org/articles/60647#video1

## The broad applicability of LightR approach

To verify that the LightR approach is applicable to other kinases, we set out to engineer LightR variants of tyrosine kinase Abl and a dual specificity kinase bRaf. Since the majority of kinases share a conserved catalytic domain structure (*Fabbro et al., 2015*), we inserted LightR switch into Abl and bRaf at a position analogous to the insertion site used in Src (*Figure 6A*), replacing K282 residue in Abl and H477 in bRaf. Our results show that illumination of LinXE cells transiently expressing LightR-Abl-GFP construct induced the phosphorylation of known Abl substrate p130Cas (*Figure 6B*). Stimulation of LightR-bRaf-Venus construct transiently expressed in LinXE cells induced phosphorylation of its direct target, MEK1, and led to downstream activation of ERK1 and ERK2 kinases at levels comparable with the constitutively active mutant of bRaf (V600E) (*Figure 6C*). To demonstrate that LightR-bRaf off-kinetics are tunable, we generated a FastLightR-bRaf variant following the same strategy used to generate FastLightR-Src. This variant exhibited significantly faster deactivation kinetics, with a half-life time around 15 min compared to approximately 3 hr half-life of LightR-bRaf (*Figure 6—figure supplement 1*). We also assessed whether FastLightR-bRaf can undergo cyclic activation and deactivation by monitoring ERK2 kinase translocation into the nucleus, a known outcome of bRaf activation (*Burack and Shaw, 2005*). Indeed, ERK2 shuttles in and out of nucleus upon activation/inactivation cycles of FastLightR-bRaf (*Figure 6D*; *Video 3*). Overall, our data show that the LightR approach can be applied to achieve light-mediated regulation of different protein kinases.

To demonstrate the broad applicability of the LightR tool to other types of enzymes, beyond kinases, we engineered LightR-Cre recombinase. Cre-recombinase has become an essential tool in biomedical research because it allows for genetic recombination and induced activation or deletion of genes (*Abremski and Hoess, 1984*; *Abremski et al., 1983*). We created four variants of Light-Cre fused to miRFP fluorescent protein that differ in their insertion site of LightR domain. The insertion sites were selected in four loops within Cre that are distant from the DNA binding site but connected to critical catalytic residues through an α-helix (*Figure 6E*). To generate these four constructs, we replaced selected amino acids (N60, D153, D189 and D278) with the LightR domain. The variant with LightR domain inserted at D153 residue in Cre showed activation in response to blue light (*Figure 6F*; *Figure 6—figure supplement 2*). This demonstrates that the LightR approach can be used broadly for the precise regulation of several types of enzymes.

## Discussion

Our study describes an optogenetic approach that provides several advantages for the interrogation of signaling pathways and demonstrates its broad applicability to address important biological questions. The key features of this method include: (1) allosteric regulation of the enzymatic activity, (2) tight temporal control of activity with tunable kinetics, (3) local regulation of activity at a subcellular level, and (4) broad applicability to different enzymes. Importantly, unlike other approaches, LightR combines all these advantages in one tool, thus, simplifying application of optogenetics in biological research.

We achieved direct regulation of enzymatic activity via an allosteric control by inserting the LightR switch into small loops within the protein structure. This modular design provides significant flexibility in the selection of the insertion site and allows for specific regulation of catalytic activity without compromising key functions of the protein such as its interactions with binding partners or its native localization in the cell (*Karginov et al., 2010*). Furthermore, variation of the flexible linkers, both between the kinase and VVD as well as between the two VVDs, or changing the insertion site for LightR in the targeted enzyme could result in reversing or altering the regulatory mechanism of

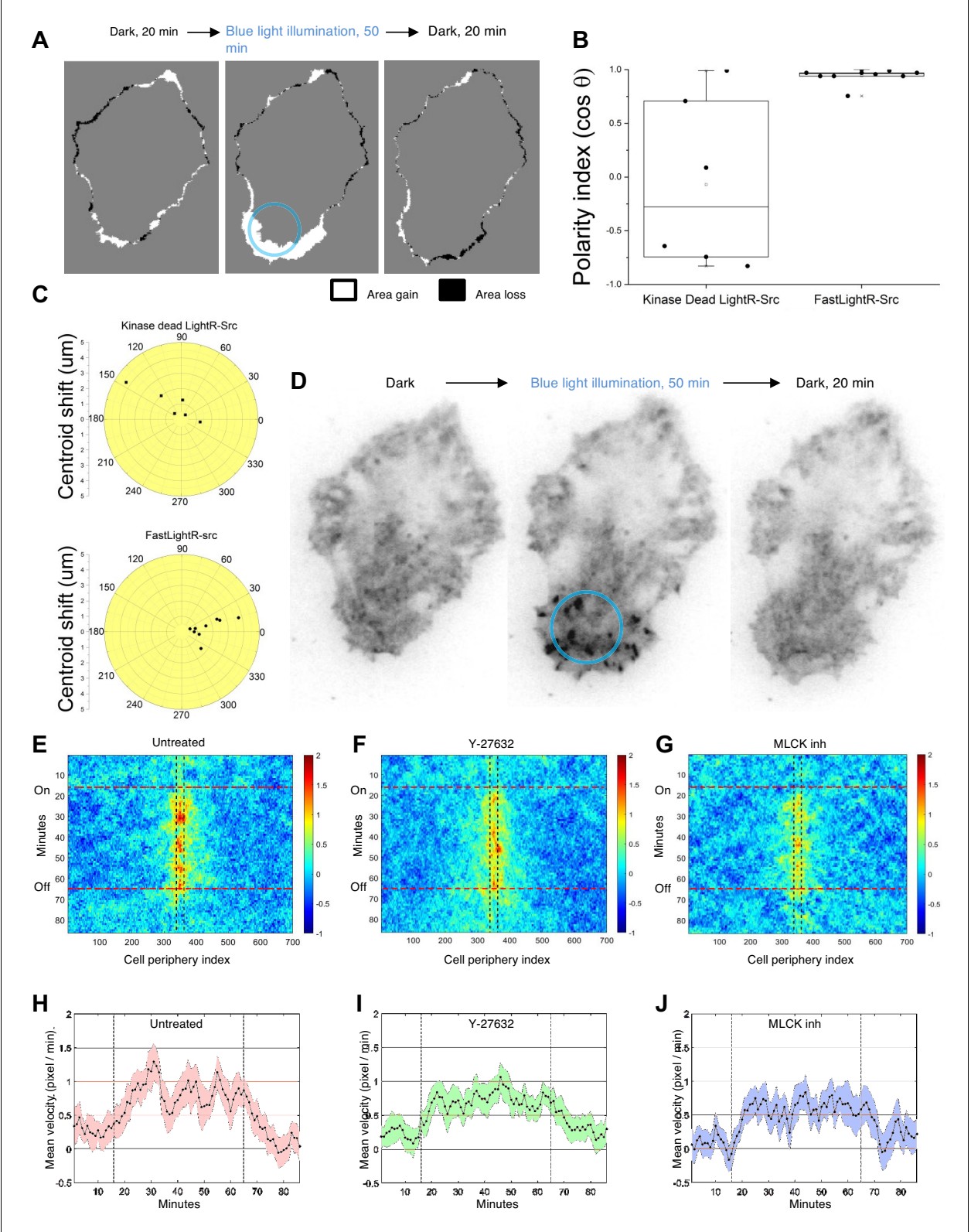

**Figure 5.** Local regulation of LightR-Src at a subcellular level. HeLa cells transiently co-expressing FastLightR-Src-mCherry-myc and Stargazin-iRFP670 (plasma membrane marker) were imaged every minute and illuminated with blue light for indicated periods of time. (**A**) Representative cell projection images showing protrusions formed between indicated time points. Blue circle outlines illuminated area. (**B**) Polarity index calculated for cells expressing catalytically inactive LightR-Src D388R (N = 6 cells) or FastLightR-Src (N = 10 cells). Box indicates the range percentile (25, 75); whiskers

*Figure 5 continued on next page*

*Figure 5 continued*

indicate the range outlier (1.5 Coef.) (**C**) Polar plot for cells in B representing the centroid's migration distance and angle of deviation (θ) relative to local blue light illumination. Black dots represent individual cells. (**D**) Inverted contrast images of FastLightR-Src-mCherry acquired at indicated time points. (**E-G**) Average cell-edge velocity kymograph for untreated cells (**E**, N = 20 cells), cells pretreated with Y-27632 (**F**, 10 μM, N = 24 cells), or MLCK Inhibitor Peptide 18 (**G**, 100 μM, N = 20 cells). Vertical lines indicate the region illuminated with blue light. Horizontal lines indicate the time of blue light illumination. (**H-J**) Mean velocity of the cell membrane region closest to the center of illumination spot. Vertical lines indicate the time of blue light illumination. Error bars represent 90% confidence interval.

The online version of this article includes the following source data and figure supplement(s) for figure 5:

**Source data 1.** Supplementary source data for *Figure 5B,C*.
**Figure supplement 1.** Local regulation of LightR-Src.
**Figure supplement 2.** Quantification of cell edge dynamics in response to local LightR-Src activation.

LightR, as was previously demonstrated for insertion of LOV2 domain in DHFR (*Reynolds et al., 2011*). While we implemented only one linker type and only one insertion site in LightR-Src, this is a venue for future work to generate more modular tools. Application of LightR approach to different classes of enzymes suggests its broad applicability for the regulation of a wide variety of protein functions in living cells. Thus, unlike other light-regulated allosteric switches (*Wu et al., 2009*; *Hongdusit et al., 2020*), this method combines multiple important advantages of optogenetic regulation with its broad applicability to a wide range of enzymes.

We showed that regulation by LightR domain is tunable, enabling different modes of regulation. LightR switch with slow off kinetics will be useful for long-term activation of LightR-enzymes; since brief periodic pulses of light will be sufficient to maintain activity while avoiding phototoxicity caused by long exposure to blue light. The FastLightR switch, on the other hand, is more suitable for studies that mimic transient, oscillatory or localized activation of a protein. It could also be used to study the kinetics of negative regulators of signaling pathways immediately after a signaling input is turned off. Activation of a LightR-enzyme requires low intensity light (*Figure 2—figure supplement 1B*, 0.6 mW/cm$^2$), minimizing the phototoxic effects of blue light illumination. This level is lower than the intensity used in other optogenetic studies using light-sensitive switches to regulate enzymes (*Dagliyan et al., 2016*; *Zhou et al., 2017*; *Wang et al., 2012*). The activation of LightR is limited to the blue light spectrum, thus enabling its multiplexing with other red-shifted optogenetic tools or FRET biosensors.

Several optogenetic approaches for the regulation of protein kinases have been described previously; however, all had specific shortcomings. Several methods regulate the localization of a kinase rather than its catalytic activity (*Kawano et al., 2015*; *Graziano et al., 2017*; *Mühlhäuser et al., 2019*; *Moitrier et al., 2019*; *Katsura et al., 2015*; *Zhang et al., 2014*). The method described by Zhou et al provides efficient control of kinases by light-regulated steric hindrance but it does not enable regulation on a subcellular level (*Zhou et al., 2017*). Furthermore, one of the regulatory domains has to be inserted in the FG loop, a substrate-interacting region in several kinases including Src family kinases (*Zhou et al., 2017*; *Shah et al., 2016*). Thus, this approach may not be applicable to some critical kinases. The insertion site that we selected for LightR domain is intentionally positioned away from any substrate binding elements, which should minimize any steric hindrance by this domain. Dagliyan et al., achieved regulation of Src through insertion of

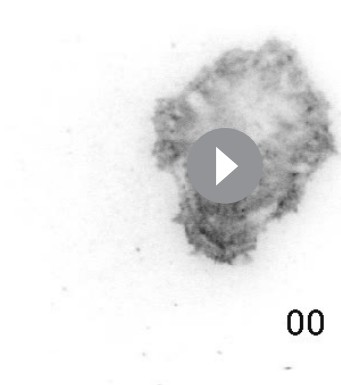

**Video 2.** Time-lapse showing HeLa cell expressing FastLightR-Src-mCherry illuminated locally with blue light (blue circle). Images were taken every minute using total internal reflection fluorescence microscopy (60X objective). Time is displayed in minutes. Images are shown as inverted contrast image representation.
https://elifesciences.org/articles/60647#video2

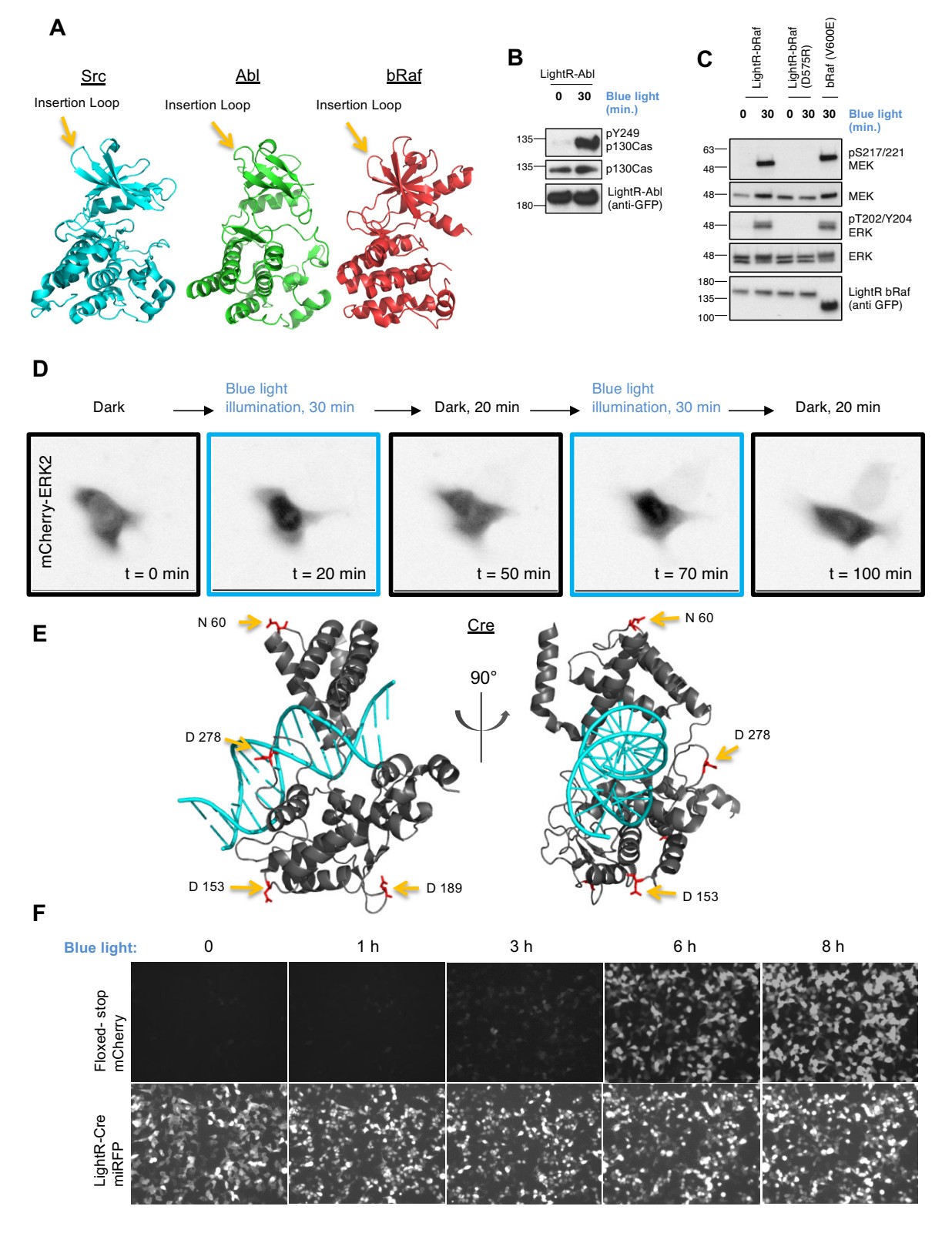

**Figure 6.** The broad applicability of LightR approach. (**A**) Crystal structures of Src (PDB 1Y57), Abl (PDB 3CS9), and bRaf (PDB 4MNF) catalytic domains. Yellow arrows indicate LightR insertion site. (**B, C**) Regulation of LightR-Abl and LightR-bRaf. LinXE cells transiently expressing GFP-LightR-Abl, LightR-bRaf-Venus, or bRaf (V600E)-Venus were exposed to continuous blue light for the specified period of time. Cell lysates were probed for the phosphorylation of indicated proteins. (**D**) LinXE cell transiently co-expressing FastLightR-bRaf-Venus and mCherry-ERK2 were imaged live while

*Figure 6 continued on next page*

*Figure 6 continued*

illuminated with blue light (as shown in the top diagram). Representative images of mCherry-ERK2 were taken at the indicated time points. (**E**) Orthogonal view of Cre recombinase structure (PDB 1MA7) indicating LightR insertion sites (yellow arrows) of the four different LightR-Cre variants. (**F**) LinXE cells transiently co-transfected with floxed-stop-mCherry reporter (upper panels) and LightR-Cre-miRFP (LightR inserted at D153, lower panels) were irradiated with pulses of blue light (2 s on 10 s off) for the indicated times. Images for all time points of each channel were acquired under the same settings and adjusted to the same brightness/contrast levels to allow for comparison of expression levels between samples. All experiments were done at least three times with similar results.

The online version of this article includes the following figure supplement(s) for figure 6:

**Figure supplement 1.** Inactivation kinetics of LightR-bRaf constructs.
**Figure supplement 2.** Analysis of LighR-Cre variants.

LOV domain at the same site that we used for the LightR approach (*Dagliyan et al., 2016*). However, this method also did not achieve local regulation at a subcellular level. Furthermore, this approach only enabled inactivation of a kinase by illumination with light. To activate a kinase at a specific time, researches have to first inhibit the kinase by keeping cells under blue light for a significant period of time, then release the inhibition by switching the light off. This requirement increases the possibility of potential cytotoxic effect of light and makes the implementation of this tool problematic for studies where activation of a kinase is desired. While the insertion of LOV domain at a different site may result in light-induced activation, as was previously demonstrated for DHFR (*Reynolds et al., 2011*), this has not been achieved for protein kinases at the moment. Thus, combined advantages of LightR enable broader application of this tool for interrogation of kinase signaling.

Tight and efficient control of LightR allowed us to mimic fast signaling dynamics of Src kinase in cells, and to establish stimulation of different signaling patterns over time. Several identified targets link early effects of Src activation to focal adhesion regulation and membrane protrusion formation. Our phosphoproteomics analysis revealed early Src-dependent phosphorylation of talin-1 at Y26 and Y70 residues (*Figure 2D*). These residues are located in the N-terminal F0 FERM domain of talin-1 (*Bouaouina et al., 2008*; *Goult et al., 2010*). When talin-1 is in the inactive closed conformation, the F0 domain remains exposed and is proposed to act as an early detector of changes in the environment (*Bouaouina et al., 2008*). Thus, phosphorylation of talin-1 on Y26 and Y70 by Src may represent an early step in the inside-out integrin activation mechanism performed by talin (*Vinogradova et al., 2002*; *Vinogradova et al., 2004*). Later phosphorylation (1–5 min) of proteins involved in formation of new focal adhesions and membrane protrusions (such as lamellipodin, p130Cas, palladin, tensin-3, cortactin, GIT1, and paxillin) correlates with the time at which we observed cell spreading and increased protrusion velocity (*Figures 4A* and *5H*). Elevated phosphorylation of many proteins at 1 hr is indicative of oncogenic transformation processes mediated by prolonged activation of Src (*Hunter and Sefton, 1980*; *Sefton et al., 1980*). Our phosphoproteomics studies also revealed that continuous Src activity induces only transient activation of MAP kinases ERK1 and ERK2 (*Figure 2—figure supplement 3*). This suggests that short term Src activation induces activation of ERK1/2 whereas prolonged activity triggers inactivation of this pathway. Previous studies demonstrated that stimulation of cells with EGF results in transient activation of ERK1 and ERK2 (*Olsen et al., 2006*; *Sasagawa et al., 2005*). In the future studies, it will be interesting

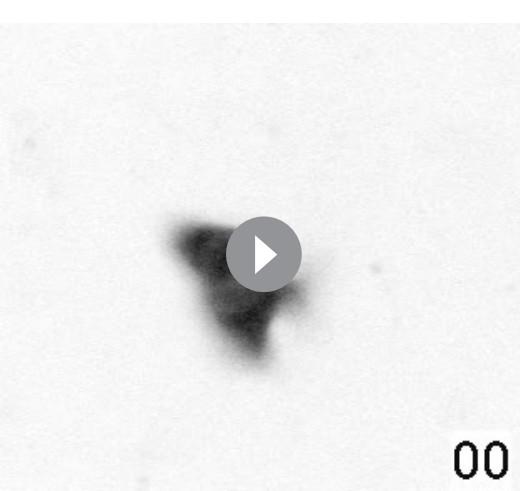

**Video 3.** Inverted contrast time-lapse images of mCherry-ERK2 expressed in LinXE cell co-expressing FastLightR-bRaf-Venus. Images were taken every minute and cell was globally stimulated with blue light (ten seconds every minute) as indicated. Time is displayed in minutes.
https://elifesciences.org/articles/60647#video3

to study whether Src also plays a role in negative regulation of ERK1/2 downstream of EGF. Importantly, our approach allowed us to identify temporal categories of Src targets; distinguishing between targets that are phosphorylated at early, intermediate, or delayed time points. Such temporal compartmentalization can be elucidated by the optogenetic approach and provides unprecedented insights into the complexity and diversity of signaling networks.

Periodic activation of kinases occurs during different cellular processes (*Kaimachnikov and Kholodenko, 2009*; *Roche et al., 1995*; *Hilioti et al., 2008*; *Jacquel et al., 2009*; *Kholodenko, 2000*; *Maeda et al., 2004*). Depending on the kinase and the physiologic context, periodicity of these cycles ranges from minutes to hours, which often determines the functional outcome (*Hilioti et al., 2008*; *Jacquel et al., 2009*; *Kholodenko, 2000*; *Maeda et al., 2004*; *Karginov et al., 2014*). Using FastLightR, we demonstrated that we can modulate activation cycles of protein kinases, Src (*Figures 3C* and *4A,C*; *Video 1*) and bRaf, (*Figure 6D*; *Video 3*) with light. Tight regulation of FastLightR allows us to control the periodicity and the amplitude of these activation pulses. Therefore, FastLightR can be potentially applied to any enzyme of interest to directly reconstruct its oscillatory signaling and identify its functional role.

The ability to regulate proteins on a subcellular level is one of the main capabilities desired in optogenetic tools. However, this has been challenging to achieve for methods employing steric hindrance and allosteric control as a mechanism of regulation (*Dagliyan et al., 2016*; *Zhou et al., 2017*). Efficiency and fast reversibility of FastLightR allowed us to control Src activity at a specific location in the cell and demonstrate that its activity is sufficient to induce local protrusion. One of the interesting observations is the fact that continuous local activation of Src induces waves of protrusions instead of causing persistent protrusions (*Figure 5E,H*). Several groups have previously described periodic protrusions at the cell edge (*Tsai and Meyer, 2012*; *Giannone et al., 2004*; *Machacek and Danuser, 2006*), and correlated this phenotype with periodic contractions due to phosphorylation of myosin light chain (MLC) (*Tsai and Meyer, 2012*; *Giannone et al., 2004*; *Machacek and Danuser, 2006*). Our results demonstrate that both kinases that phosphorylate MLC (ROCK and MLCK) participate in the formation of the recurrent waves of protrusion in response to local Src activation. However, ROCK appears to be less important for maintaining the overall elevated protrusion velocity downstream of Src activation (*Figure 5—figure supplement 2B*). Inhibition of MLCK resulted in much more noticeable reduction in average protrusion velocity (*Figure 5—figure supplement 2B*). Previous studies suggested that MLCK predominantly acts at the cell periphery where it mediates formation of stable protrusions and persistent cell migration (*Totsukawa et al., 2004*). Our data is consistent with these prior studies and suggest that MLCK is a predominant factor regulating cell protrusions downstream of Src.

Several previous studies described approaches for the regulation of Cre recombinase activity by light (*Wu et al., 2020*; *Morikawa et al., 2020*; *Kawano et al., 2016*; *Taslimi et al., 2016*; *Sheets et al., 2020*; *Kennedy et al., 2010*). However, these methods are designed based on the split protein reassembly strategy, which has its limitations. Split proteins tend to reassemble spontaneously, leading to undesirable basal activity of engineered split Cre (*Dagliyan et al., 2018*; *Liu and Tucker, 2017*). Moreover, when activated, not all split halves of the protein reassemble, which often leads to suboptimal activation levels of the engineered enzyme (*Dagliyan et al., 2018*). Furthermore, this method requires co-expression of two proteins and, to achieve optimal results, both split halves must be co-expressed in equimolar ratios in a cell. LightR overcomes some of these limitations, because LightR-Cre is, by design, a single construct that expresses a complete Cre enzyme fused with LightR domain. Although high levels of LightR-Cre expression sometimes lead to low basal activity (*Figure 6—figure supplement 2*), this could be solved by optimizing the expression level of LightR-Cre and/or by modifying the linkers connecting LightR domain to the Cre recombinase. We believe that such optimization is feasible because we did not detect any basal activity for the LightR-kinases regardless of their expression level in cells.

In summary, the LightR-approach to regulate enzyme activity demonstrates versatility and broad applicability to a variety of enzymes and can thus help unravel signaling pathways and networks as well as study defined biological processes that are direct consequences of the regulated enzymes.

## Materials and methods

### Antibodies, chemical reagents, materials, and cell lines

The following antibodies were used: anti-phospho-p130Cas (Y249) (BD Pharmigen cat. no. 558401), anti-p130Cas (BD Pharmigen cat. no. 610271), anti-GFP (Clontech, cat. no. 632381), anti-paxillin (Fisher Scientific, cat. no. BDB612405), anti-phospho-paxillin (Y118) (Invitrogen, cat. no. 44–722G), anti-Src (Santa Cruz, cat. no. 8056), anti-GAPDH (Ambion, cat. no. AM4300), anti-Myc (Millipore, cat. no. 05–724), anti-phospho-MEK1/2 (Ser17/221) (Cell Signaling, cat. no. 9121), anti-MEK1/2 (Cell Signaling, cat. no. 9122), anti-p44/42 (ERK1/2) (Cell Signaling, cat. no. 9102), anti-p44/42 (ERK1/2) (Thr202/Tyr204) (Cell Signaling, cat. no. 9101).

The reagents used were: IgG-coupled agarose beads (Millipore, cat. no. IP04-1.5ML), Leupeptin hemisulfate (Gold Biotechnology, cat. no. L-010–5), Aprotinin (Gold Biotechnology, cat. no. A-655–25), Y-27632 dihydrochloride (Millipore-Sigma, cat. no. Y0503), MLCK Inhibitor Peptide 18 (Cayman, cat. no. 224579-74-2), 2X Laemmli Sample buffer (BIO-RAD, cat. no. 161–0737), 2-Mercaptoethanol (Fisher Chemical, cat. no. 60-24-2), Trypsin (Promega, cat. no. V5113), Trypsin-digested BSA for coating tubes (Sigma, cat. no. A7906), Fetal Bovine Serum (Omega Scientific, cat. no. FB-01). Coverslips for live imaging were purchased from (ThermoFisher, cat. no. 25CIR-1.5).

The materials used were: C18 cartridges (Waters, cat. no. WAT02350), TMT11plex (ThermoFisher, cat. no. A34807), TMT10plex (ThermoFisher, cat. no. 90406), Fe-NTA spin columns (ThermoFisher, cat. no. A32992), glutathione sepharose (GE Healthcare, cat. no. GE17-0756-01).

The following cell lines were used: HeLa cells (ATCC, cat. No. CCL- 2), Human embryonic kidney HEK293T cells (ATCC, cat. no. CRL-3216), and human embryonic kidney LinXE cell line (derived from HEK 293 cells, gift from Klaus Hahn Lab, UNC) (*Bravo-Cordero et al., 2013*). All cell lines were cultured at 37°C and 5% carbon dioxide in DMEM medium supplemented with 10% FBS and 1 mM L-glutamine. All experiments were performed with cells grown for less than 20 passages after thawing. All cells were tested negative for *Mycoplasma* contamination. Cell lines identity was confirmed by the supplier using STR analysis. DH5α bacteria cells (NEB, cat. No. C2987H) were used for GST-paxillinN-C3 production.

### Molecular dynamics modeling

All LightR-Src structures used for molecular dynamics simulations were created using the Src catalytic domain from PDB: 1Y57 combined with dark and lit state VVD from PDB: 2PD7 and 3RH8, respectively. Using UCSF Chimera (version 1.12) software (*Pettersen et al., 2004*), the asymmetric flexible linkers, GPGGSGG and GSGGPG, were respectively added to the N- and C- termini of VVD and the monomers were linked with a 22 amino acid flexible linker (GGS)$_4$G(GGS)$_3$. These VVD constructs were then inserted into the active Src catalytic domain using Chimera's structure editor. The flexible linkers were then refined using Modeller (*Sali and Blundell, 1993*). All structures were energy minimized in NAMD (version 2.13) using a steepest descent gradient for 1000 steps (*Phillips et al., 2005*). These structures were then equilibrated for 0.5 ns at increasing temperature from 60 K to 310 K, then for an additional 1 ns at 310 K, with generalized Born implicit solvent with an ion concentration of 0.15 M. All MD simulations were then carried out under these same conditions over 100 ns. Simulations for the LightR constructs were conducted in triplicate for a total simulation time of 300 ns per structure. Time points for the pre-equilibrated structure, before RMSD convergence, were discarded for analysis to ensure structure used was reasonable. Starting structures were qualitatively validated by evaluating RMSD convergence (*Figure 1—figure supplement 1A*). The RMSD analysis and visualization was carried out using VMD (version 1.9.3) and principal component analysis (PCA) was conducted using the NMWiz GUI for ProDY (*Bakan et al., 2011*). Distances between the N and C-termini of the LightR domain as well as the length of the flexible linker when extended in the dark state were determined using the average position of these residues from the MD simulations. The averaged structure of the dark and lit state LightR-Src were opened in Chimera to obtain these distances with Chimera's measurement tool.

Amino acid sequence of LightR domain is:

*GPGGSGG*HTLYAPGGYDIMGYLIQIMNRPNPQVELGPVDTSCALIL CDLKQKDTPIVYASEAFLYMTGYSNAEVLGRNCRFLQSPDGMVKP KSTRKYVDSNTINTMRKAIDRNAEVQVEVVNFKKNGQRFVNFLTM

**IPVRDETGEYRYSMGFQCETE**GGSGGSGGSGGSGGGSGGSGGS
HTLYAPGGYDIMGYLIQIMNRPNPQVELGPVDTSCALILCDLKQK
DTPIVYASEAFLYMTGYSNAEVLGRNCRFLQSPDGMVKPKSTRK
YVDSNTINTMRKAIDRNAEVQVEVVNFKKNGQRFVNFLTMIPVR
DETGEYRYSMGFQCETE*GSGGPG*.

Linkers are italicized. Bolded and underlined sequences are the first and second VVD sequences, respectively.

## Molecular biology

LightR gene was codon optimized so that the two tandem VVD DNA sequences are as different as possible to make cloning using PCR easier. LightR sequence was designed so that the two VVD proteins were connected with a flexible twenty-two amino acid linker $(GGS)_4G(GGS)_3$. LightR DNA sequence was ordered as a gBlock from Integrated DNA Technologies. The gBlock was amplified using PCR with forward primer encoding a `GPGGSGG` linker and a 24–28 nucleotide sequence that anneals upstream of the insertion site of interest. The reverse primer encodes a `GSGGPG` linker and 24–28 nucleotide sequence that anneals downstream of the insertion site of interest. The resulting PCR product of this reaction acts as a megaprimer that we use to insert LightR domain at a site of interest in a gene using a modification of QuickChange site-directed mutagenesis (*Karginov and Hahn, 2011*). LightR-Src-mCherry-myc construct was generated by using RapR-Src-mCherry-myc construct (*Klomp et al., 2016*) and replacing iFKBP insert with the LightR. In the resulting construct, LightR is replacing G288 in cSrc (position in avian Src). Constitutively active CA-Src-mCherry-myc construct was generated by replacing Cerulean in CA-Src-Cerulean-myc construct (*Klomp et al., 2016*) with mCherry using a modification of QuickChange site-directed mutagenesis (*Karginov and Hahn, 2011*). Stargazin-iRFP670 was generated from the previously described stargazin-mCherry construct (*Karginov et al., 2014*), by replacing mCherry with iRFP670 using a modification of QuickChange site-directed mutagenesis (*Karginov and Hahn, 2011*). pmiRFP670-N1 was a gift from Vladislav Verkhusha (Addgene plasmid # 79987; http://n2t.net/addgene:79987; RRID:Addgene_79987) (*Shcherbakova et al., 2016*). pCAG-iCre was a gift from Wilson Wong (Addgene plasmid # 89573; http://n2t.net/addgene:89573; RRID:Addgene_89573), and pcDNA3.1_Floxed-STOP-mCherry was a gift from Moritoshi Sato (Addgene plasmid # 122963; http://n2t.net/addgene:122963; RRID:Addgene_122963). Using a modification of QuickChange site-directed mutagenesis (*Karginov and Hahn, 2011*), iCre gene was cloned into the pmiRFP670-N1 backbone to obtain iCre-miRFP670 plasmid. Similarly, ERK2 gene from pCEFL-ERK2 (a gift from Dr. Channing Der's lab, UNC) was cloned into mCherry-C1 backbone to obtain mCherry-ERK2 plasmid. We obtained GFP-Abl (*Homo sapiens*) construct (gift from Dr. Steven Dudek, UIC) and introduced P242E/P249E mutations to make it constitutively active using site-directed mutagenesis, bRaf-Venus construct bearing V600E mutation was a gift from Dr. John O'Bryan (MUSC), LeGO-iV2 was a gift from Boris Fehse (Addgene plasmid # 27344; http://n2t.net/addgene:27344; RRID:Addgene_27344) (*Weber et al., 2008*), pMD2.G was a gift from Didier Trono (Addgene plasmid # 12259; http://n2t.net/addgene:12259; RRID:Addgene_12259), psPAX2 was a gift from Didier Trono (Addgene plasmid # 12260; http://n2t.net/addgene:12260; RRID:Addgene_12260), GST-paxillinN-C3 construct was a gift from Dr. Michael Schaller, WVU (*Lyons et al., 2001*).

## Expression of engineered constructs

All DNA constructs were transfected transiently using Fugene 6 (Promega Corporation) transfection reagent according to the manufacturer protocol.

## GST-paxillin purification

N-terminal fragment of paxillin was purified following previously described procedure (*Lyons et al., 2001*). Briefly, GST-paxillinN-C3 construct was expressed in DH5α bacteria cells following induction with 0.1 mM Isopropyl β-D-1-thiogalactopyranoside for 4 hr. Bacterial pellet was resuspended in 30 ml of TETN buffer (20 mM TRIS pH 8, 100 mM NaCl, 1 mM EDTA, 0.5% Triton X100) and lysed by sonication. GST-paxillinN-C3 was purified from the cleared lysates by affinity chromatography using Glutathione Sepharose following previously described protocol (*Lyons et al., 2001*).

### *In vitro* kinase assay

A detailed protocol for this experiment was previously described (*Cai et al., 2008*). Briefly, Src kinase constructs bearing an mCherry and a myc tandem tags at the C-terminus were transiently overexpressed in LinXE cells. Cells were exposed to continuous blue light (3 mW/cm$^2$, 465 nm wavelength) for the indicated times or kept in the dark. Lysates were then collected under red light illumination (to prevent activation of LightR) using the lysis buffer (20 mM HEPES-KOH, pH 7.8, 50 mM KCl, 1 mM EGTA, 1% NP40, 1 mM NaF, 0.2 mM Na3VO4, aprotinin 16 µg/ml, and Leupeptin hemisulfate 3.2 µg/mL). Lysates were centrifuged at 4000 rpm, 4°C, for 10 min, and the cleared lysates were incubated with ProteinG sepharose beads conjugated with the anti-myc antibody (4A6 from Millipore-Sigma) for 1.5 hr at 4°C. Beads were then washed with wash buffer (20 mM Hepes-KOH, pH 7.8, 100 mM NaCl, 50 mM KCl, 1 nM EGTA, 1% NP40) and then with kinase reaction buffer (25 mM HEPES, pH 7.5, 5 M MgCl$_2$, 0.5 mM EGTA, 0.005% BRIJ-35). The beads were resuspended in kinase reaction buffer and incubated with 0.1 mM ATP and 0.05 mg/ml purified N-terminal fragment of paxillin (GST-paxillinN-C3) at 37°C for 10 min. The reaction was terminated by adding 2X Laemmli sample buffer with 5% v/v 2-Mercaptoethanol then incubating at 110°C for 5 min. The phosphorylation of paxillin was examined by western blotting.

### Biochemical characterization of LightR kinases

LinXE cells grown in 3 cm plates to 60–80% confluency were transfected with the LightR construct of interest using Fugene six transfection reagent as recommended by the manufacturer. Transfected cells were incubated overnight at 37°C in the dark to prevent unwanted activation. The following day, the cells were exposed to blue light by placing them 10 cm above an HQRP LED Plant Grow Panel Lamp System (3 mW/cm$^2$, 465 nm wavelength). Temperature was maintained during incubation times by placing the setup inside a tissue culture incubator (*Figure 2—figure supplement 1A*). Light was shining continuously for the desired time. In experiments that required cycles of activation and inactivation, we manually unplugged the LED panel during inactivation times and plugged it back in during activation times before we lysed the cells. At the end of the experiment time course, media was aspirated and cells were lysed under safe red-lights with 700 µl of 2X Laemmli sample buffer containing 5% v/v 2-Mercaptoethanol. Cell lysates were collected and incubated at 110°C for 5 min. 10–15 µL of the cell lysates were analyzed by a western blot to probe for phosphorylation of endogenous proteins.

### Characterization of LightR-Cre

To assess LightR-Cre activity we used the previously described Floxed-STOP-mCherry reporter system (*Kawano et al., 2016*). LinXE cells were co-transfected overnight with LightR-Cre-iRFP670 and Floxed-STOP-mCherry DNA constructs (1:9 ratio) and kept in the dark. The next day, cells were pulsed with light using the HQRP LED Plant Grow Panel Lamp System (3 mW/cm$^2$, 465 nm wavelength). A microcontroller (Arduino Uno) and power relay (IoT Relay, Digital Data Loggers INC.) were used to turn on the LED panel for 2 s every 10 s. Cells were then imaged using EVOS Auto 2 Invitrogen fluorescence microscope using 20 × air objective to analyze the expression of mCherry reporter.

### Production of LightR-Src HeLa stable cell line

Using LighR-Src-mCherry-myc plasmid as a template, LightR-Src-mCherry-myc gene was PCR-amplified with primers that introduced restriction sites NotI and BsiWI on the 5' and 3' end, respectively. This PCR product and the LeGO-iV2 plasmid were digested with NotI and BsrGI and then ligated to generate a LightR-Src-mCherry-myc lentiviral construct. This construct was then co-transfected with pMD2.G, a plasmid expressing VSV-G lentivirus envelope protein, and psPAX2, a second-generation lentiviral packaging plasmid, into HEK 293 T cells (ATCC CRL-3216). After 1–3 days, the conditioned media was centrifuged (1000 × g, 10 min, 25°C), and virus-containing supernatant was used to directly infect HeLa cells. Transduced HeLa cells were sorted via FACS by selecting the brightest 20 percentile of mCherry-expressing cells.

## Mass spectrometry

### Sample preparation

Urea lysates were prepared as previously described (*Dittmann et al., 2019*). Briefly, lysate was reduced with 10 mM dithiothreitol (DTT), alkylated with 55 mM iodoacetamide, and digested with trypsin overnight. Digested peptides were subjected to desalting using C18 cartridges (Waters), and lyophilized. Peptides were labeled with TMT 11plex isobaric mass tags and stored at −80 C.

### Phosphotyrosine enrichment and mass spectrometry analysis

TMT-labeled peptide samples were subjected to a previously described two-step enrichment process (*Dittmann et al., 2019*), with the following modifications: First, 12 µg of 'super' 4G10 (*Mou et al., 2018*) was utilized in place of the commercial antibody from Millipore Sigma. Second, peptides were eluted from antibody-conjugated beads twice with 0.2% trifluoroacetic acid in milliQ water. Eluted peptides were directly added to immobilized metal affinity chromatography (IMAC) Fe-NTA spin columns (ThermoFisher Scientific). Fe-NTA columns were used according to the manufacturer's instructions. Phosphopeptides were eluted twice with 20 µL of elution buffer into a BSA-coated 1.5 mL tube, and eluted peptides were dried down in a speed vacuum concentrator. Dried peptides were resuspended in 10 µL of 5% acetonitrile in 0.1% formic acid and loaded directly onto an in-house packed analytical capillary column (50 µm ID ×12 cm, 5 µm C18) with an integrated electrospray tip (1–2 µm orifice). Eluates were then subjected to LC-MS/MS as previously described with the following modifications (*Dittmann et al., 2019*): (1) The mass spectrometer was operated with a spray voltage of 2.5 kV. (2) Selected ions were HCD fragmented at normalized collision energy 32%. (3) MS/MS acquisition was performed at a resolution of 60,000. Limited LC-MS/MS analysis of the most abundant peptides to adjust for channel-to-channel loading variation was carried out on an Orbitrap Q-Exactive Plus mass spectrometer using ~15 ng of peptide. Supernatant was loaded onto an acidified trapping column and analyzed with gradients as follows: 0–13% solvent B in 4 min, 13–42% in 46 min, 42–60% in 7 min, 60–100% in 3 min, and 100% for 8 min, before equilibrating back to Solvent A. Full scans (MS1) were acquired in the m/z range of 350–2000 at a resolution of 70,000 (m/z 100). The top 10 most intense precursor ions were selected and isolated with an isolation width of 0.4 m/z. Selected ions were HCD fragmented at normalized collision energy (NCE) 33% at a resolution of 70,000.

### Peptide identification and quantification

Raw mass spectral data files were processed as previously described (*Dittmann et al., 2019*) with the following changes: (1) Raw files were processed with Proteome Discoverer version 2.2.0.388 (ThermoFisher Scientific) and searched against the human SwissProt database using Mascot version 2.4.1 (Matrix Science). (2) Peptide spectrum matches (PSMs) for phosphopeptides were filtered for ion score ≥20 and precursor isolation interference (<35%), and PSMs for the most abundant peptides in IP supernatant runs were filtered for ion score ≥25.

### Data analysis

Data from three independent mass spectrometry runs was transformed as previously described using IP supernatants run on the Q-Exactive Plus (*Dittmann et al., 2019*). For each phosphopeptide, relative quantification was represented as a ratio between TMT ion intensities from each time point (10 s, 30 s, 1 min, 5 min, 60 min) and the 0 s LightR-Src condition. Data were filtered for peptides that were common to each of three runs.

### Visualization and statistical analysis

To identify phosphopeptides with significant changes in abundance relative to the 0 s LightR-Src condition, we utilized paired Student's t-test (p-value<0.05). Additionally, to ensure phosphopeptide changes were not simply background, we filtered for peptides with average abundances (n = 3) that were >1.4 relative to the 0 s LightR-Src condition. The average abundance for each phosphopeptide was log2-transformed and visualized using MATLAB (version R2019b, Bioinformatics Toolbox version 4.13, MathWorks). Data were plotted with the 'clustergram' function with hierarchical clustering using Euclidean distance.

## PCA and STRING network analysis

For PCA, all phosphoproteomic data were normalized to basal abundances in LightR-Src cells (0 s LightR-Src HeLa). Phosphopeptide abundances were averaged across all three biological replicates, then log2-transformed. Abundances were re-centered around 0 by subtracting the mean of each phosphopeptide across all conditions. PCA was performed using Scikit-learn (v0.19.1) in Python (v3.6.0). For STRING network analysis of principal component 1, loading scores from PCA were extracted for all phosphopeptides, rank-ordered (decreasing), and the gene IDs of all phosphopeptides with loading score >0.05 were provided as input to STRING (v11.0) (*Szklarczyk et al., 2019*). Disconnected nodes were hidden from the network representation. Source code for PCA of phosphoproteomics data can be found on GitHub (https://github.com/flowerc/LightR-Src).

## Cell imaging and image analysis

### Sample preparation and imaging hardware

On the day of imaging, HeLa cells transiently expressing desired construct were seeded 3 hr before imaging at 30–40% confluency on a glass coverslip (0.16–0.19 mm thick) that was coated overnight with fibronectin (5 mg/L). In experiments using inhibitors, cells were pretreated with the appropriate inhibitor 2 hr before imaging. Live cell imaging was performed in Leibovitz (L-15) imaging medium supplemented with 5% fetal bovine serum 2–4 hr after seeding HeLa cells on coverslips. Cells were kept at 37 C using an open heated chamber (Warner Instruments) and imaged using Olympus IX-83 microscope controlled by Metamorph software and equipped with Xcite 120 LED (Lumen Dynamics) light source, objective-based total internal reflection fluorescence (TIRF) system, 445 nm and 561 nm laser lines, and Image EMX2 CCD camera (Hamamatsu). Imaging for analysis of Src-induced cell spreading and bRaf-mediated translocation of Erk2 was conducted using Olympus UPlanSAPO 40× (oil, N.A. 1.25) objective. Imaging of Src local activation effects and its localization to focal adhesions was performed using PlanApo N 60 × TIRF objective (oil, NA 1.45).

### Cell spreading analysis

Using epifluorescence imaging, we selected cells co-expressing LightR-Src-mCherry and stargazin-iRFP670 (plasma membrane marker) (*Chen et al., 2000*). Time-lapse images were acquired every minute. To induce cell spreading, cells were globally illuminated with pulses of blue light (50 ms pulse every second, 50 pulses every minute). Stargazin-iRFP670 images were used to create a binary mask based on intensity thresholding. The binary mask was generated by MovThresh (Matlab 2019b version) software package (*Tsygankov et al., 2014*). Then using Metamorph, we calculated cell area. To measure the area change for each cell, we divided the cell area at any given time by the average area of the same cell prior to blue light irradiation. Finally, the average and 90% confidence interval were calculated for each time point of all cells treated in similar conditions.

### Local blue light stimulation

A focused beam of 445 nm laser light was delivered to local area at the cell periphery using Olympus Cell-TIRF module in FRAP-mode. The light was fixed throughout the stimulation period and was irradiated for fifty-four seconds every minute.

### Polarity index calculation

Polarity index calculation was adopted from a previously described method (*Wu et al., 2009*). Briefly, we used Metamorph software to calculate the angle of deviation (θ) of the cell's centroid from the blue light in response to local blue light stimulation (*Figure 5—figure supplement 1A*). To obtain the polarity index, we calculated cos θ.

### Centroid shift

The centroid shift distance (*Figure 5—figure supplement 1A*) was determined using Metamorph by measuring the number of pixels the centroid moved in response to local blue light and multiplying this value by the pixel size (0.4 µm for a 40X objective) on our camera. Centroid shift (µm)=number of pixels x 0.4.

### Quantification of cell edge dynamics

In order to quantify the response of cell edge to the local kinase activation with light, we performed the following steps of the analysis:

### Cell edge extraction

Each cell in the time-lapse records was segmented to obtain the binary cell mask at every time frame. The standard pixel wise tracing of an object boundary was applied to extract the cell outline. Then, the outlines were processed with the CellGeo2016 software package (*Tsygankov et al., 2014*) to smooth the edge by removing all fine-scale (slender) protrusions like filopodia and retraction fibers. In addition, the boundaries were resampled with a large number of points (N = 700) to achieve the consistency of the boundary representations between different cells and different time points.

### Velocity calculation

The velocity of each point representing cell edge at time $t$ was calculated as the distance to the nearest point of the cell edge at time $t + t$ divided by $t$. In this study, we used $t = 5$ minutes.

### Velocity kymographs

In order to represent the cell edge dynamics over the recording time as a single graph, we constricted kymographs by stacking the velocity values along the cell edge sequentially from the first (top) to the last (bottom) time frame. The resulting data matrix was visualized as a color map with the highest positive velocity (protrusion) in red and the highest negative velocity (retraction) in blue. Although we use the same number of points in the cell boundary representation, the perimeter of these boundaries varies from frame to frame, which can create artificial horizontal misalignment depending on how the first point of the boundary is chosen. To avoid this issue at the edge region of our interest, we used our automated algorithm (see below) to align velocity values such that the edge position along the line of the most active protrusion near the site of the light activation was in the middle of the kymograph. This way, any boundary misalignments due to the variation of the cell perimeter can occur only on the sides of the kymographs, where our further comparative quantification was not performed. Code used for this analysis is publicly available at https://github.com/aaz-hur/LightR.

### Alignment of kymograph rows

Most of the processing steps of the kymograph construction algorithm was adopted from the EdgeProps software (*Zhurikhina, 2018*). The only difference is the method of finding the point on the cell outline at each time frame to be positioned in the middle column of the kymograph. In this method (*Figure 5—figure supplement 2A*), we first apply Gaussian Filter to the cell outline to obtain a smoothed representation of the cell edge at times $t$ and $t + t$. Then, we measure the distance from each point on the boundary at time $t$ to the boundary at time $t + t$ along the directions of the normal vectors to the boundary at time $t$. The local maximum of this measure in the vicinity of the laser activation is taken as the middle position in the velocity kymograph. The calculation of this local maximum is repeated by the algorithm for each time frame so that all boundaries are consistently aligned over the whole-time record. For MovThresh we used Matlab 2019b. For CellGeo2016 we used Matlab 2016b. For EdgeProps we used Matlab 2017b. For Kymograph construction and analysis, we used Matlab 2018b, Matlab 2019b and Matlab 2020a. All versions are compatible with Matlab 2020a. Code used for this analysis is publicly available at https://github.com/aazhur/LightR.

### Comparative analysis of the light-induced edge movement

Once the edge velocity from different time frames is properly centered in the kymographs, we can compare the response of different cell types by averaging the kymographs of all cells in a given phenotypic group. Furthermore, by choosing a central stripe in the kymographs (the area between the vertical dash lines in *Figure 5E–H*) and averaging it along the perimeter, we convert the spatiotemporal representation of the protrusive cell response at the vicinity of the stimulation site into the velocity versus time plots shown in *Figure 5H–J*. In this study, we use the stripe width of 42 pixels, which corresponds to arc length of 11.2 μm. Code used for this analysis is publicly available at

https://github.com/aazhur/LightR. Plots in *Figure 5H–J* were generated using R 3.6.3 and then R4.0.0 R studio 1.2.1335, all of which are compatible with the latest version.

### Statistical analysis of the light-induced edge movement

After converting the spatiotemporal response of the cell edge to the light activation into the velocity versus time plots as described above, we identified the average velocity value for each cell.

To evaluate statistical significance of the difference in these measures between cell phenotypes, we used Wilcoxon test, the nonparametric equivalent of the two-sample t-test. *Figure 5—figure supplement 2B* was generated using R 3.6.3 and then R4.0.0 R studio 1.2.1335, all of which are compatible with the latest version.

### DNA sequences for LightR and LightR constructs

See *Supplementary file 1*.

## Acknowledgements

We would like to thank Dr. Susan Barrett, senior research imaging specialist at Olympus Corporation of the Americas, and Dr. Peter Toth, director of the Fluorescence Imaging Core at UIC, for their technical help with the imaging experiments, as well as Dr. Jennifer Klomp for her helpful insights during the pursuit of this project. We are grateful to the NIH and Chicago Biomedical Consortium for the funding (CBC Pilot Grant, R21CA212907, R21CA159179, and R01GM118582 grants to AVK; R21CA223915 to AVK and JR; training grant T32 HL007829-22 to MS, JF, and MB; and P01 HL060678 to AVK, VN and JR). We are also thankful to NCI for the funding (CA210180 and CA238720 grants to FMW) and for the U.S. Army Research Office (ARO) (W911NF-17-1-0395 to DT).

## Additional information

### Funding

| Funder | Grant reference number | Author |
| --- | --- | --- |
| Chicago Biomedical Consortium | | Andrei V Karginov |
| Army Research Office | W911NF-17-1-0395 | Denis Tsygankov |
| National Institutes of Health | R21CA159179 | Andrei V Karginov |
| National Institutes of Health | R01GM118582 | Andrei V Karginov |
| National Institutes of Health | R21CA223915 | Jalees Rehman<br>Andrei V Karginov |
| National Institutes of Health | HL007829-22 | Mark Shaaya<br>Jordan Fauser<br>Martin Brennan |
| National Institutes of Health | P01 HL060678 | Viswanathan Natarajan<br>Jalees Rehman<br>Andrei V Karginov |
| National Institutes of Health | CA210180 | Forest M White |
| National Institutes of Health | CA238720 | Forest M White |

The funders had no role in study design, data collection and interpretation, or the decision to submit the work for publication.

### Author contributions

Mark Shaaya, Formal analysis, Validation, Investigation, Visualization, Methodology, Writing - original draft; Jordan Fauser, Formal analysis, Investigation, Visualization, Methodology, Writing - review and editing; Anastasia Zhurikhina, Software, Formal analysis, Investigation, Methodology; Jason E Conage-Pough, Formal analysis, Investigation, Methodology, Writing - original draft; Vincent Huyot, Investigation, Methodology; Martin Brennan, Jacob Matsche, Shahzeb Khan, Investigation; Cameron

T Flower, Formal analysis, Investigation; Viswanathan Natarajan, Forest M White, Supervision, Funding acquisition, Writing - review and editing; Jalees Rehman, Funding acquisition, Writing - review and editing; Pradeep Kota, Formal analysis, Supervision, Funding acquisition, Investigation, Writing - review and editing; Denis Tsygankov, Formal analysis, Supervision, Funding acquisition, Methodology; Andrei V Karginov, Conceptualization, Supervision, Funding acquisition, Project administration, Writing - review and editing

## Author ORCIDs
Mark Shaaya ⓘ https://orcid.org/0000-0003-4066-2693
Jordan Fauser ⓘ https://orcid.org/0000-0002-0675-8977
Jason E Conage-Pough ⓘ http://orcid.org/0000-0002-1614-9374
Cameron T Flower ⓘ http://orcid.org/0000-0002-9632-9913
Jalees Rehman ⓘ https://orcid.org/0000-0002-2787-9292
Forest M White ⓘ http://orcid.org/0000-0002-1545-1651
Denis Tsygankov ⓘ http://orcid.org/0000-0002-1180-3584
Andrei V Karginov ⓘ https://orcid.org/0000-0003-2370-6383

## Decision letter and Author response
Decision letter https://doi.org/10.7554/eLife.60647.sa1
Author response https://doi.org/10.7554/eLife.60647.sa2

## Additional files

### Supplementary files
• Supplementary file 1. DNA sequences for LightR and LightR constructs.

• Transparent reporting form

### Data availability

The raw mass spectrometry data and associated tables have been deposited to the ProteomeXchange Consortium via the PRIDE partner repository with the dataset identifier: PXD018162. All data generated or analyzed during this study are included in the manuscript and supporting files.

The following dataset was generated:

| Author(s) | Year | Dataset title | Dataset URL | Database and Identifier |
|---|---|---|---|---|
| Conage-Pough JE | 2020 | Optogenetic Src Temporal Signaling | https://www.ebi.ac.uk/pride/archive/projects/PXD018162 | PRIDE, PXD018162 |

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
