## [Decision Letter]

**Acceptance summary:**

This paper presents an original and potent optogenetic switch that allows allosteric control of enzyme activity. Albeit the use of fungal-based LOV domains to orthogonally control different cellular processes is not new, this light-regulated allosteric switch presents unique features that makes of LightR a powerful tool to dissect and explore basic and fundamental biological questions. The dataset presents fine examples of how Src activity can be controlled with the switch of a light, allowing tunable, temporal and spatial control of its activity (with subcellular resolution), revealing important details about its dynamics. The modular features of LightR are clearly exemplified in its use not only to modulate Src activity (ON in the light/ OFF in the dark), but also in the control of the kinases Abl and bRaf, as well as of Cre recombinase. For sure LightR (and future variations of this type of optogenetic allosteric switch) will continue to enlighten not only aspects of kinase biology, but also unforeseen features of diverse cellular processes.

**Decision letter after peer review:**

Thank you for submitting your article "LightR: A Light-Regulated Allosteric Switch Enables Temporal and Subcellular Control of Enzyme Activity" for consideration by *eLife*. Your article has been reviewed by three peer reviewers, including Luis F Larrondo as the Reviewing Editor and Reviewer #1, and the evaluation has been overseen by Jonathan Cooper as the Senior Editor.

The reviewers have discussed the reviews with one another and the Reviewing Editor has drafted this decision to help you prepare a revised submission.

Summary:

The work by Shaaya et al. describes a new strategy for engineering allosteric regulation of proteins by blue light, by genetically inserting a light-sensitive clamping domain (LightR) into a loop of an enzyme's surface. The authors present examples of this strategy on three different well-studied kinases (Src, Abl, and bRaf) and on Cre recombinase. LightR is based on a tandem VVD (LOV domain) fusion and works as a clamp-like switch that opens in the dark and closes (because of VVD-VVD intramolecular dimerization) in response to blue-light. This rationally designed light-sensitive domain, based on a LOV fungal domain, constitutes a nice addition to the repertoire of optogenetic switches described in past years and exemplifies an interesting strategy to achieve intramolecular optogenetic control.

The ability to orthogonally regulate by light any enzyme that can be inactivated by inserting the LightR domain into a surface loop is novel and welcome, and the LightR variant with faster off-kinetics is a useful addition. As shown by the authors, this allows achieving reversible and repeatable light activation at different time scale with subcellular resolution.

The data described by the authors indicate that this strategy could represent a broadly applicable approach.

Essential revisions:

The major concern relates to some parts of the manuscript that require clarification. There are aspects of how the data are discussed, and how the methods are described, that ought to be improved in order to help colleagues grasp key aspects of the work and value its significance and reach. It is also important to contextualize the results in order to understand the novelty and contributions of the presented strategy.

1) The MD simulation methods used are not sufficiently detailed. Moreover, the scale of the molecular dynamics work presented in Figure 1 does not contribute to the claims made by the authors. More specifically, it relies on single short simulations of a homology model constructed with little experimental knowledge of the actual orientation of the VVD domain relative to the kinase catalytic domain.

2) Some experimental detail on the expression of LightR constructs in cells is unclear or lacking. From the Materials and methods section, it appears that most of the work was done in LinXE cells transiently expressing LightR constructs and that the phosphoproteomics work was done in HeLa cells stably expressing the constructs, but this is unclear in the main text. Also, the specifics of the LightR-kinase and LightR-Cre recombinase constructs are unclear: the figures seem to indicate that LightR-Src and FastLightR-Src are myc-tagged, while LightR-Abl and bRaf are GFP-tagged, but this is not mentioned in the main text or the Materials and methods. The authors should briefly describe the major components of the constructs used when they first appear in the main text and must also provide the full sequences of the constructs in the supplement.

3) Little validation data is presented for either transient or stable expression of the constructs: only single-time-point western blots for transient LightR-kinase expression in Figures 2, 3 and 6, no data for transient LightR-Cre expression and no data for stable LightR-Src expression. This data likely already exists and should probably be presented in supplementary figures.

4) The experiments using ROCK and MLCK inhibitors are problematic and may not support the conclusion that ROCK is less important a mediator of Src's effect on cell-edge velocity than MLCK. Y-27632, the ROCK inhibitor used, is reported to have an IC50 of 140 nM, so the authors' choice of a 10-µM working concentration is fairly reasonable, but the MLCK inhibitor peptide is reported to have an IC50 of 50 nM and to be cell-permeable, so the authors' choice of a 100-µM working concentration is worryingly high and may have off-target effects that are actually mediating the observed greater decrease in cell-edge velocity. The authors need to justify the concentrations used or otherwise justify the relative importance of ROCK and MLCK for mediating Src's effect on membrane protrusion. This is a relatively minor portion of this paper, which focuses on the development of the LightR tool, and removing this claim would not significantly affect the strength of the paper.

5) The authors propose that the novelty of the method lies in a combination of features: 1) regulation is achieved allosterically (rather than a sterically or by localization/sequestration) 2) regulation is achieved with subcellular precision and 3) the regulation shows tunable kinetics. However, each of these properties have been established elsewhere and here the main contribution lies in the combination of all three. The tunable kinetics in this work made use of a previously characterized mutation and thus isn't particularly unique. Other work has shown precise spatiotemporal localization (e.g. Cavanaugh et al. (2020) Cell Protoc Cell Biol 86:e102), but by localization rather than allostery. And allosteric regulation of an enzyme by light is shown in both (Dagliyan et al., 2016, and Lee et al. (2008) Science 322:438), but as the authors point for Dagliyan et al. "it does not achieve local control". Thus, the novelty of the current work requires a *very* precise definition of what is new.

Likewise, the work is most similar to (Dagliyan et al., 2016). Both manuscripts describe the regulation of Src by a LOV domain, and the authors write that the main distinction between this work and their own is that Dagliyan et al. "does not achieve local control and triggers inactivation rather than activation of the protein". The directionality of the regulation (activating vs. deactivating) is not necessarily fundamental – others have observed both activation and deactivation by LOV2 insertion, even within the same protein (Reynolds et al., 2011). These aspects need to be explained/discussed.

6) Modulation of CRE by LightR is not mentioned in the Discussion. Moreover, there have been several reports where CRE activity (split strategy) has been utilized to control its activity (PMID: 32709899; PMID: 32358538; PMID: 27723747; PMID: 27065233, including work where CRE is regulated by VVD dimerization (e.g. Sheets et al., 2020 and Kennedy et al., 2010). Those examples should be referenced, and the advantage of having a light-activated CRE in only one construct, as opposed to two (split strategy) should be mentioned.

7) The work does not give a good sense of whether (or not) insertion of the tandem VVD domain perturbs kinase function. For example, the domain insertions in (Lee et al. (2008) Science 322:438) result in a dramatic decrease in the WT enzyme activity. The authors might address this by comparing their in vivo measurements to existing data for activated native Src or bRaf. Another method may be using *in vitro* kinase assays to directly measure activity in lit and dark states. If these experiments cannot be currently conducted, the authors should at least clearly discuss this point.

8) "This indicates that the open conformation of LightR clamp in the dark can cause changes in the catalytic domain of Src that will be reversed upon illumination with blue light". The authors should also discuss if this approach could be implemented (by changing the type of linker, or where LightR is inserted) such that the "distorted" conformation can be reached in the light state, whereas the "WT/active" one could be achieved in the dark.

The description of the LightR should briefly mention the size of the utilized VVD (compared to what has been used in optogenetic systems), and the type of linker that was added in between both VVDs. This info is extractable from the figures, but it would help if it is explicitly mentioned: for example, how many different GS linker sizes were simulated in silico, and how many were experimentally tried before defining the definitive LightR construct.

---

## [Author Response]

Essential revisions:The major concern relates to some parts of the manuscript that require clarification. There are aspects of how the data are discussed, and how the methods are described, that ought to be improved in order to help colleagues grasp key aspects of the work and value its significance and reach. It is also important to contextualize the results in order to understand the novelty and contributions of the presented strategy.1) The MD simulation methods used are not sufficiently detailed. Moreover, the scale of the molecular dynamics work presented in Figure 1 does not contribute to the claims made by the authors. More specifically, it relies on single short simulations of a homology model constructed with little experimental knowledge of the actual orientation of the VVD domain relative to the kinase catalytic domain.

We apologize for not providing enough details about the MD simulations. We included more details in the revised manuscript under the “Development of a Light-Regulated Kinase” subsection in the Results section, and under the “Molecular dynamics modeling” subsection in the Materials and methods section.

We had previously failed to mention that the data presented were the combined result of three independent 100 ns simulations. In Figure 1—figure supplement 1, we also included a plot of the RMSD of the system over the course of the simulation as validation of our homology model. Convergence of the RMSD of the system is indicative of a structure occupying an energetic minima and thus represents a reasonable structure for analysis. While we acknowledge that there are intrinsic limitations to homology modeling, it provides a method to guide protein engineering efforts and evaluate possible mechanisms of regulation before experimental data are obtained. Similar approaches have been used in previously published work to assess engineered molecules and propose potential mechanism of action

(https://pubmed.ncbi.nlm.nih.gov/27980211/, https://doi.org/10.1007/s10989-018-9791-9, https://doi.org/10.1007/s10989-006-9058-8, https://doi.org/10.3389/fmolb.2020.00004, https://pubmed.ncbi.nlm.nih.gov/26673131/).

2) Some experimental detail on the expression of LightR constructs in cells is unclear or lacking. From the methods section, it appears that most of the work was done in LinXE cells transiently expressing LightR constructs and that the phosphoproteomics work was done in HeLa cells stably expressing the constructs, but this is unclear in the main text. Also, the specifics of the LightR-kinase and LightR-Cre recombinase constructs are unclear: the figures seem to indicate that LightR-Src and FastLightR-Src are myc-tagged, while LightR-Abl and bRaf are GFP-tagged, but this is not mentioned in the main text or the Materials and methods. The authors should briefly describe the major components of the constructs used when they first appear in the main text and must also provide the full sequences of the constructs in the supplement.

We appreciate this comment. In the revised manuscript, we specified the cell lines used in every experiment in the main text and figure legends (Figures 2-6; Figure 2—figure supplements 1-4; Figure 3—figure supplement 1; Figure 5—figure supplement 1B; and Figure 6—figure supplements 1, 2). We also specified whether the constructs are transiently or stably expressed. Furthermore, we described the major components of the constructs when they first appear in the main text and provided their full DNA sequence at the end of the Materials and methods section.

3) Little validation data is presented for either transient or stable expression of the constructs: only single-time-point western blots for transient LightR-kinase expression in Figures 2, 3 and 6, no data for transient LightR-Cre expression and no data for stable LightR-Src expression. This data likely already exists and should probably be presented in supplementary figures.

We apologize for not providing validation data for the HeLa stable cell line (LightR-Src HeLa) we generated. We included a western blot in the revised manuscript in Figure 2—figure supplement 1C that shows comparable expression levels of LightR-Src and endogenous Src.

We also provided western blots showing transient expression levels of LightR-Src,

LightR-Abl, and LightR-bRaf at each time point. These can be found in Figure 2C;

Figure 3A-C; Figure 6B, C; Figure 2—figure supplement 1B; Figure 3—figure supplement 1; Figure 6—figure supplement 1.

To compare LightR-Cre-iRFP expression levels between different time points, we provided images of LightR-Cre-iRFP in Figure 6F and in Figure 6—figure supplement 2. In the previous submission, we failed to indicate that all images were taken using the same illumination and acquisition settings and were adjusted to the same brightness/contrast level; so the expression levels can be compared between samples. Even though LightR-Cre-iRFP was not measured by Western blot, fluorescence microscopy suggests that similar expression levels were achieved in different experiments. This is now clarified in the revised version.

4) The experiments using ROCK and MLCK inhibitors are problematic and may not support the conclusion that ROCK is less important a mediator of Src's effect on cell-edge velocity than MLCK. Y-27632, the ROCK inhibitor used, is reported to have an IC50 of 140 nM, so the authors' choice of a 10-µM working concentration is fairly reasonable, but the MLCK inhibitor peptide is reported to have an IC50 of 50 nM and to be cell-permeable, so the authors' choice of a 100-µM working concentration is worryingly high and may have off-target effects that are actually mediating the observed greater decrease in cell-edge velocity. The authors need to justify the concentrations used or otherwise justify the relative importance of ROCK and MLCK for mediating Src's effect on membrane protrusion. This is a relatively minor portion of this paper, which focuses on the development of the LightR tool, and removing this claim would not significantly affect the strength of the paper.

We understand the concern about the concentration we used. We decided to use 100 μm of the MLCK inhibitor peptide based on the published work referenced on the manufacturer’s website (https://faseb.onlinelibrary.wiley.com/doi/full/10.1096/fj.09-146118#fsb2fj09146118-fig-0005). In this work, authors pre-treated microglia with 100 µM MLCK inhibitor to achieve inactivation. Furthermore, a separate study demonstrated that MKCK inhibitor peptide 18 maintains high specificity for up to 4,000 fold of its IC50, which is a concentration of 200 μm (J. Med. Chem. 42(5), 910-919 (1999)).

5) The authors propose that the novelty of the method lies in a combination of features: 1) regulation is achieved allosterically (rather than a sterically or by localization/sequestration) 2) regulation is achieved with subcellular precision and 3) the regulation shows tunable kinetics. However, each of these properties have been established elsewhere and here the main contribution lies in the combination of all three. The tunable kinetics in this work made use of a previously characterized mutation and thus isn't particularly unique. Other work has shown precise spatiotemporal localization (e.g. Cavanaugh et al. (2020) Cell Protoc Cell Biol 86:e102), but by localization rather than allostery. And allosteric regulation of an enzyme by light is shown in both (Dagliyan et al., 2016, and Lee et al. (2008) Science 322:438), but as the authors point for Dagliyan et al. "it does not achieve local control". Thus, the novelty of the current work requires a *very* precise definition of what is new.Likewise, the work is most similar to (Dagliyan et al., 2016). Both manuscripts describe the regulation of Src by a LOV domain, and the authors write that the main distinction between this work and their own is that Dagliyan et al. "does not achieve local control and triggers inactivation rather than activation of the protein". The directionality of the regulation (activating vs. deactivating) is not necessarily fundamental – others have observed both activation and deactivation by LOV2 insertion, even within the same protein (Reynolds et al., 2011). These aspects need to be explained/discussed.

We added clarifications in the main text that better explain the advantage of the tool. The main contribution of this tool, specified in the Introduction, is that, unlike previous approaches for allosteric regulation, it combines all advantages of optogenetics (including regulation at a subcellular resolution) and enables broad applicability. We also further discussed the advantages of LightR as an allosteric activation tool in the Discussion section and dedicated a paragraph to comparing this method to other optogenetic tools enabling direct regulation of protein kinase activity.

6) Modulation of CRE by LightR is not mentioned in the Discussion. Moreover, there have been several reports where CRE activity (split strategy) has been utilized to control its activity (PMID: 32709899; PMID: 32358538; PMID: 27723747; PMID: 27065233, including work where CRE is regulated by VVD dimerization (e.g. Sheets et al., 2020 and Kennedy et al., 2010). Those examples should be referenced, and the advantage of having a light-activated CRE in only one construct, as opposed to two (split strategy) should be mentioned.

To address this comment, we dedicated a paragraph in the Discussion section of the revised manuscript to compare LightR-Cre with other split optogenetic Cre tools.

7) The work does not give a good sense of whether (or not) insertion of the tandem VVD domain perturbs kinase function. For example, the domain insertions in (Lee et al. (2008) Science 322:438) result in a dramatic decrease in the WT enzyme activity. The authors might address this by comparing their in vivo measurements to existing data for activated native Src or bRaf. Another method may be using in vitro kinase assays to directly measure activity in lit and dark states. If these experiments cannot be currently conducted, the authors should at least clearly discuss this point.

This is a great point. To address this, we added a panel in Figure 2A that shows results of an in vitro kinase assay for LightR-Src and constitutively active Src. These results demonstrate that the insertion of LightR domain in Src kinase completely abolishes its activity in the dark and restores it in the light to levels comparable to that of constitutively active Src.

8) "This indicates that the open conformation of LightR clamp in the dark can cause changes in the catalytic domain of Src that will be reversed upon illumination with blue light". The authors should also discuss if this approach could be implemented (by changing the type of linker, or where LightR is inserted) such that the "distorted" conformation can be reached in the light state, whereas the "WT/active" one could be achieved in the dark.The description of the LightR should briefly mention the size of the utilized VVD (compared to what has been used in optogenetic systems), and the type of linker that was added in between both VVDs. This info is extractable from the figures, but it would help if it is explicitly mentioned: for example, how many different GS linker sizes were simulated in silico, and how many were experimentally tried before defining the definitive LightR construct.

To address this comment, we discussed a possibility to change the directionality of LightR regulation by changing the linkers or the insertion site of LightR (Discussion section).

We also indicated the size of LightR domain (335 amino acid total size) and specified the linker that was used between VVD domains. We simulated and tested only one linker. The linkers connecting LightR domain to the catalytic domain were selected based on previous studies (PMID: 20581846 and PMID: 24609359).

The GS-repeat linker between VVD domains was selected to provide sufficient length (approximately 25-30 Å) to allow dissociation and association of VVD monomers. This is now clarified in the revised version.